# A review of the effectiveness of operational curtailment for reducing bat fatalities at terrestrial wind farms in North America

**Evan M. Adams** *, **Julia Gulka, Kathryn A. Williams**

Biodiversity Research Institute, Portland, Maine, United States of America

* evan.adams@briwildlife.org

**Data Availability Statement:** The data used in this study were obtained from a combination of publicly available and anonymous curtailment studies. The project name of public studies and the source we

## Abstract

Curtailment of turbine operations during low wind conditions has become an operational minimization tactic to reduce bat mortality at terrestrial wind energy facilities. Site-specific studies have demonstrated that bat activity is higher during lower wind speeds and that operational curtailment can effectively reduce fatalities. However, the exact nature of the relationship between curtailment cut-in speed and bat fatality reduction remains unclear. To evaluate the efficacy of differing curtailment regimes in reducing bat fatalities, we examined data from turbine curtailment experiments in the United States and Canada in a meta-analysis framework. We used multiple statistical models to explore possible linear and non-linear relationships between turbine cut-in speed and bat fatality. Because the overall sample size for this meta-analysis was small (n = 36 control-treatment studies from 17 wind farms), we conducted a power analysis to assess the number of control-treatment curtailment studies needed to understand the relationship between fatality reduction and change in cut-in speed. We also identified the characteristics of individual curtailment field studies that may influence their power to detect fatality reductions, and in turn, contribute to future meta-analyses. We found strong evidence that implementing turbine curtailment reduces fatality rates of bats at wind farms; the estimated fatality ratio across all studies was 0.37 (p < 0.001), or a 63% decrease in fatalities. However, the nature of the relationship between the magnitude of treatment and reduction in fatalities was more difficult to assess. Models that represented the response ratio as a continuous variable (e.g., with a linear relationship between the change in cut-in speed and fatalities) and a categorical variable (to allow for possible non-linearity in this relationship) both had substantial support when compared using $AIC_c$. The linear model represented the best fit, likely due to model simplicity, but the non-linear model was the most likely without accounting for parsimony and suggested fatality rates decreased when the difference in curtailment cut-in speeds was 2m/s or larger. The power analyses showed that the power to detect effects in the meta-analysis was low if fatality reductions were less than 50%, which suggests that smaller increases in cut-in speed (i.e., between different treatment categories) may not be easily detectable with the current dataset. While curtailment is an effective operational mitigation measure overall, additional well-designed curtailment studies are needed to determine precisely whether higher cut-in speeds can further reduce bat fatalities.

used to obtain each study are listed in Table 1 of the article (the minimum data required to recreate the meta-analysis, also found as a .csv file in S3 Table). Data were compiled from their original reports by the authors of the curtailment study review within the CanWEA report (2018) and by the American Wind Wildlife Institute (AWWI) in the American Wind Wildlife Information Center (AWWIC) program. The AWWIC summarizes bird and bat data collected at 227 operating U.S. wind energy projects (more details are found at: https://awwi.org/resources/awwic-bat-technical-report/). The data sourced from AWWIC were supplied to the authors in response to the submission and acceptance of a detailed proposal to the Wind Wildlife Research Fund request for proposals. In order to request data used in this article, researchers will need to submit a detailed proposal to AWWI at info@awwi.org or respond to a Wind Wildlife Research Fund request for proposals. AWWIC is a collaborative of independent data owners, managed by AWWI, therefore data usage proposals will be evaluated and approved by all data owners.

**Funding:** This study was funded by the Wind Wildlife Research Fund (https://awwi.org/wind-wildlife-research-fund/).

**Competing interests:** The authors have declared that no competing interests exist.

## Introduction

Wind energy development is increasing rapidly worldwide, and hundreds of thousands of bat fatalities are estimated to occur per year due to collisions with terrestrial wind energy facilities in North America [1–3]. Turbine attraction is a leading hypothesis for high observed fatality rates, particularly in migratory tree bats [4, 5]. Between 70% and 80% of bats killed at wind energy facilities in the U.S. are migratory tree bats, including hoary bat (*Lasiurus cinereus*), eastern red bat (*L. borealis*), and silver-haired bat (*Lasionycteris noctivangans;* [3, 6–8]). While fatality rates vary among sites, the estimated cumulative mortality is high enough to be considered a serious conservation concern for at least one North American bat species [9, 10].

Operational curtailment, also known as "blanket curtailment," is a mitigation approach that involves raising the threshold for ambient wind speed ("cut-in speed") at which turbines begin generating electricity. Curtailment of turbine operations during low wind conditions, particularly in late summer and fall when bat fatality rates are highest, is an operational minimization tactic to reduce bat fatality at terrestrial wind facilities [11]. Below the cut-in speed, turbine blades can spin with the wind but do so much more slowly, especially if blades are "feathered," or pitched to catch as little wind as possible. Because bats tend to be more active at lower wind speeds, increasing turbine cut-in speed can significantly reduce bat fatality [1, 6]. However, myriad other factors also influence fatality risk (e.g., time of year, weather, turbine dimensions, and landscape characteristics; [8]), and a great deal of variability has been reported in the level of fatality reduction achieved by curtailment [12]. The relationship between the magnitude of increased cut-in speed (e.g., the amount by which the cut-in speed of turbines is adjusted from the factory settings) and the degree of fatality reduction in bats remains unclear.

Curtailment based on wind speed and time of day/year has operational and financial implications for wind facility operators [13]. At present, the trade-off between turbine energy production and bat fatality minimization is poorly understood. Larger increases in cut-in speeds will further reduce power generation, but the implications for fatality reduction are less clear. Fatality monitoring is subject to uncertainty introduced by biased observation processes and small sample sizes, and there is currently limited evidence that raising cut-in speeds above 4.5 m/s will further reduce bat fatalities [12]. A synthesis of the available data from existing curtailment studies will better quantify the relative benefits of increasing turbine cut-in speed for reducing bat collision fatality.

Meta-analysis provides a method to account for multiple types of uncertainty and use predictor variables to explain patterns across studies [14]. Using this approach, we aimed to evaluate the effectiveness of curtailment regimes in reducing bat fatalities at terrestrial wind projects in North America. We identified three objectives: 1) evaluate existing control-treatment curtailment study data for bats in a meta-analysis to examine the relative benefit of increased curtailment cut-in speeds; 2) assess the power of the a meta-analysis to quantify fatality reduction using data simulation; and 3) understand how different site or survey characteristics (e.g., fatality rates, study length, and carcass persistence) influence the power of individual curtailment studies to detect a difference in bat fatality rates between control and treatment groups. In combination, these objectives help to identify 1) the effect of curtailment strategies on bat fatality reduction, 2) how much information is needed to be certain of these effects, and 3) how to design curtailment experiments to maximize the value of their results in contributing to our overall understanding of the relationship between cut-in speeds and bat fatalities.

## Methods

To assess objective 1, we used a *meta-analysis* to examine the effect of changing cut-in speeds on bat fatality rates using a response ratio approach. As we did not have a predetermined

assumption about the nature of the relationship between fatality rate and the change in cut-in speed between control and treatment, we used a model selection framework that included both continuous and categorical models to describe for changes in cut-in speed to compare the evidence for linear and non-linear relationships with fatality reductions. By including the cut-in speed of the control group in the model set, the absolute cut-in speed was also allowed to influence fatality rates. We also examined how covariates like study location and turbine dimensions could help explain the relationship between fatality rate and change in cut-in speed (see *Meta-analysis* section for additional information on model building and selection).

For objectives 2 and 3, we assessed the likelihood that the above *meta-analysis* would provide statistically significant results and determined the number of control-treatment pairs needed to be confident in our understanding of the relationship between fatality rate and change in cut-in speed. Thus, we conducted two power analyses. The first power analysis (the "*meta-analysis power analysis*") was designed to quantify the power of the meta-analysis under different hypothetical scenarios about the relationship between fatality rate and change in turbine cut-in speed. The second (the "*fatality estimation power analysis*") was designed to inform future curtailment studies and fatality monitoring efforts at operating wind energy facilities. All analyses were conducted in R [15], and all analysis scripts were documented in the (S1 Appendix).

## Data inclusion

Data for this analysis went through identification and screening processes prior to inclusion (Fig 1). Data were collected in part from the American Wind Wildlife Information Center (AWWIC, Accessed in August 2019), which compiles private and public data from post-construction fatality monitoring studies at individual wind energy projects in the United States (n = 26) [7]. Here we define a project as an experiment happening at a given site and year (i.e., a unique combination of site and year), while a study is a given control-treatment pair (multiple studies can occur at one project). Data from several additional projects in the U.S. and Canada were harvested from publicly available reports, specifically a 2018 report from the Canadian Wind Energy Association (CanWEA; n = 21, with n = 9 duplicates with AWWIC projects) [12]. Curtailment treatments based solely on ambient wind speed were of primary interest for this analysis. Projects were included in the analysis if both a treatment and control group with fatality estimates at different cut-in speeds were present at the same project site. Data were excluded from analysis if there was no change in cut-in speed between treatments (e.g., testing other fatality reduction methods) or no measurement of treatment effect (e.g., no control treatment). The remaining projects (n = 26) were conducted at 17 wind energy project sites in the U.S. and Canada from 2005–2016. There were instances where multiple experimental cut-in speeds were tested simultaneously at the same site, such that multiple studies from the same project and year shared a control, resulting in a total of 36 control-treatment pairs (hereafter 'studies'; Table 1).

Fatality estimates, which were reported from the original studies, had already been adjusted for detection probability (observer ability to detect carcasses that are present) and carcass persistence (rate of removal of carcasses by scavengers) using searcher efficiency trials and carcass persistence trials, respectively [16]. There are multiple approaches for correcting fatality estimates that differ in their assumptions accounting for detection error resulting from carcass removal and searcher efficiency [16, 17]. Studies in this analysis primarily used the Huso and Shoenfeld estimators [18, 19]. However, some studies used the Erickson estimator [20], MNRF estimator [21], or custom calculations to adjust fatality estimates for carcass persistence and searcher efficiency. An adjusted bat fatality estimate per turbine for the treatment and control

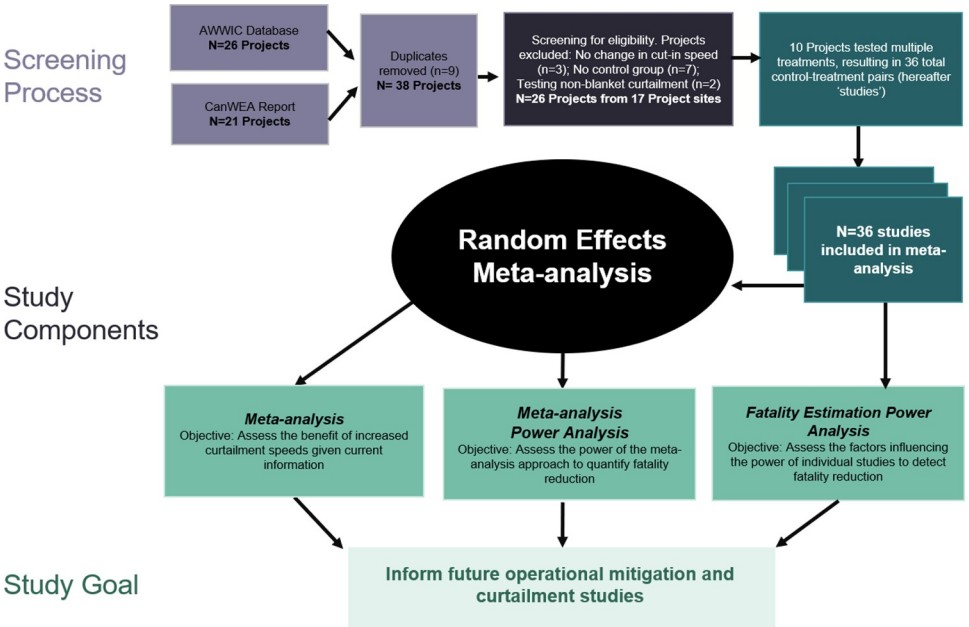

**Fig 1. Meta-analysis data flow diagram for the study based on the PRISMA guidelines.** After compiling all projects in the American Wind Wildlife Information Center (AWWIC) database and the Canadian Wind Energy Association (CanWEA) 2018 synthesis that reported turbine curtailment, we removed duplicates between the two sources. Individual studies within those projects were determined to be suitable for analysis if they used a curtailment treatment based solely on ambient wind speed (e.g., no other environmental variables) and multiple cut-in speeds that could be compared. These studies were then used within a meta-analysis framework to examine study objectives.

were presented with upper and lower 90–95% confidence intervals (CIs) in a majority of studies (n = 26), while some presented bat fatalities per turbine (n = 3) or percent decrease (n = 7) without uncertainty estimates.

Study periods varied somewhat between individual curtailment studies, with some studies examining specific time periods throughout the night or focusing on different windows of time during fall migration. Fatality estimates were converted to bat fatalities per turbine-hour by dividing fatalities per turbine by the total number of curtailment hours. Few studies reported species-specific fatality rates, so fatality estimates were for all bat species combined. Turbine cut-in speeds varied among control and treatment groups, and differences between the control-treatment pairs required standardization. In most studies, the control group's cut-in speed was 3.5 m/s (a common cut-in speed set by turbine manufacturers), though values ranged from 3.0 to 5.0 m/s. Experimental cut-in speeds varied from 4.0 to 7.0 m/s. Due to this variation and the small sample size of available studies, the change in cut-in speed between treatment and control (Δ cut-in speed) was used as the treatment magnitude.

## Meta-analysis

We used a random effects meta-analysis to determine the efficacy of curtailment strategies for fatality reduction while accounting for uncertainty in each study's effect size estimate [22]. The effect size of each study was calculated as a log-transformed ratio between the estimated fatality of the treatment and the control, both in the unit of bat fatalities per turbine hour (i.e., the log-transformed response ratio, hereafter 'RR,' [23]). In instances where only a percent decrease was reported, RR was calculated as log(1-(% decrease/100). Confidence intervals (90% or 95% depending on the estimator used) for control and treatment fatality estimates were log-

**Table 1. Bat fatality data from curtailment studies in the AWWIC database and CanWEA 2018 report.**

| Project Site | Year | Region | RD | Cut-in Speed Cont. | Cut-in Speed Exp. | Cut-in Speed Δ | Effect Fatal. Ratio | Effect % | Source |
|---|---|---|---|---|---|---|---|---|---|
| Anonymous East | 2014 | East | 77 | 3.5 | 4.5 | 1.0 | 0.50 ± 0.16 | 50 | AWWIC |
| Anonymous East | 2015 | East | 77 | 3.5 | 5.5 | 2.0 | 1.00 ± 0.34 | 0 | AWWIC |
| Anonymous East | 2015 | East | 77 | 3.0 | 4.0 | 1.0 | 0.91 ± 0.30 | 9 | AWWIC |
| Anonymous East | 2016 | East | 77 | 3.0 | 5.0 | 2.0 | 0.69 ± 0.28 | 31 | AWWIC |
| Anonymous East | 2016 | East | 77 | 3.0 | 6.0 | 3.0 | 0.53 ± 0.25 | 47 | AWWIC |
| Casselman Wind | 2008 | East | 77 | 3.5 | 5.0 | 1.5 | 0.13 ± 0.13 | 87 | AWWIC |
| Casselman Wind | 2008 | East | 77 | 3.5 | 6.5 | 3.0 | 0.26 ± 0.17 | 74 | AWWIC |
| Casselman Wind | 2009 | East | 77 | 3.5 | 5.0 | 1.5 | 0.32 ± 0.16 | 68 | AWWIC |
| Casselman Wind | 2009 | East | 77 | 3.5 | 6.5 | 3.0 | 0.24 ± 0.13 | 76 | AWWIC |
| Criterion | 2012 | East | 93 | 4.0 | 5.0 | 1.0 | 0.38 ± 0.14 | 62 | AWWIC |
| Laurel Mountain | 2011 | East | 82 | 3.5 | 4.5 | 1.0 | 0.42 ± 0.15 | 58 | AWWIC |
| Pinnacle Wind Force | 2012 | East | 95 | 3.0 | 5.0 | 2.0 | 0.53 ± 0.15 | 47 | AWWIC |
| Pinnacle Wind Force | 2013 | East | 95 | 3.0 | 5.0 | 2.0 | 0.42 ± 0.18 | 58 | AWWIC |
| Pinnacle Wind Force | 2013 | East | 95 | 3.0 | 6.5 | 3.5 | 0.25 ± 0.12 | 75 | AWWIC |
| Anonymous Midwest | 2010 | Midwest/West | 82 | 3.5 | 4.8 | 1.3 | 0.53 ± 0.19* | 47 | CanWEA |
| Anonymous Midwest | 2010 | Midwest/West | 82 | 3.5 | 4.0 | 0.5 | 0.28 ± 0.13* | 72 | CanWEA |
| Anonymous Pac. SW | 2012 | Midwest/West | 101 | 3.5 | 4.8 | 1.3 | 0.80 ± 0.26* | 20 | CanWEA |
| Anonymous Pac. SW | 2012 | Midwest/West | 101 | 3.5 | 4.0 | 0.5 | 0.65 ± 0.22* | 35 | CanWEA |
| Anonymous Pac. SW | 2012 | Midwest/West | 101 | 3.5 | 4.8 | 1.3 | 0.62 ± 0.20* | 38 | CanWEA |
| Fowler Ridge 1 | 2010 | Midwest/West | 89 | 3.5 | 5.0 | 1.5 | 0.50 ± 0.11 | 50 | AWWIC |
| Fowler Ridge 1 | 2010 | Midwest/West | 89 | 3.5 | 6.5 | 3.0 | 0.21 ± 0.06 | 79 | AWWIC |
| Fowler Ridge 1 | 2011 | Midwest/West | 89 | 3.5 | 4.5 | 1.0 | 0.64 ± 0.23* | 36 | AWWIC |
| Fowler Ridge 1 | 2011 | Midwest/West | 89 | 3.5 | 5.5 | 2.0 | 0.38 ± 0.17* | 62 | AWWIC |
| Fowler Ridge 1 | 2012 | Midwest/West | 89 | 3.5 | 5.0 | 1.5 | 0.16 ± 0.06 | 84 | AWWIC |
| Lakefield | 2016 | Midwest/West | 77 | 3.5 | 5.0 | 1.5 | 0.56 ± 0.34 | 44 | AWWIC |
| Summerview | 2005 | Midwest/West | 80 | 4.0 | 7.0 | 3.0 | 0.61 ± 0.22* | 39 | CanWEA |
| Summerview | 2007 | Midwest/West | 80 | 4.0 | 5.5 | 1.5 | 0.94 ± 0.26 | 6 | CanWEA |
| Wild Cat 1 | 2013–15 | Midwest/West | 100 | 5.0 | 7.0 | 2.0 | 0.20 ± 0.07 | 80 | AWWIC |
| Wild Cat 1 | 2016 | Midwest/West | 100 | 5.0 | 6.9 | 1.9 | 0.41 ± 0.33 | 59 | AWWIC |
| Bull Hill | 2013 | Northeast | 100 | 3.0 | 5.0 | 2.0 | 0.70 ± 0.23 | 30 | AWWIC |
| Enbridge Wind | 2012 | Northeast | 82 | 3.5 | 5.5 | 2.0 | 0.38 ± 0.18* | 62 | CanWEA |
| Raleigh Wind | 2014 | Northeast | 77 | 3.5 | 4.5 | 1.0 | 0.23 ± 0.05 | 77 | CanWEA |
| Sheffield | 2012 | Northeast | 94 | 4.0 | 6.0 | 2.0 | 0.37 ± 0.13 | 63 | AWWIC |
| Talbot Wind | 2013 | Northeast | 101 | 3.5 | 5.5 | 2.0 | 0.04 ± 0.09* | 96 | CanWEA |
| Wolfe Island | 2011 | Northeast | 93 | 3.2 | 4.5 | 1.3 | 0.52 ± 0.20* | 48 | CanWEA |
| Wolfe Island | 2011 | Northeast | 93 | 3.2 | 5.5 | 2.3 | 0.40 ± 0.17* | 60 | CanWEA |

Information provided includes project name, year, geographic region, and rotor diameter in meters (RD); control cut-in speed (Cont.), experimental cut-in speed (Exp.), and change in cut-in speed (Δ), all in m/s; and treatment effect information, including the mean fatality ratio ± SE (Fatal. Ratio) and percent decrease in fatality between treatments (%). Studies from the same project and year were tested simultaneously and shared a control. Some studies lacked information on fatality uncertainty (*); for these, the mean SE from the multiple imputation process is shown.

transformed and converted into separate standard error (SE) estimates, then combined into a single SE estimate for each study's fatality ratio using the delta method [23, 24]. Some studies lacked effect size uncertainty estimates (n = 12); to account for our uncertainty in the SEs from these studies, we used a multiple imputation approach [25, 26] using the *mice* R package [27].

Study SEs were estimated using correlations with other known variables (effect size, $\Delta$ cut-in speed, project site, number of experimental turbines, number of study nights, and rotor diameter) using predictive mean matching [27]. Multiple imputation (k = 50) was used to incorporate the uncertainty in this process into the meta-analysis results. For each imputation, a covariance matrix was built for all studies based on the observed and estimated SEs, where covariance was estimated between two studies when they shared a control. The covariance matrix was then used for weighting the studies in the meta-analysis.

To conduct the meta-analysis quantifying the relationships between $\Delta$ cut-in speed and bat fatality reduction, we used the 'rma.mv' function from the *meta* R package [28] and ran nine candidate models. The primary explanatory variable was $\Delta$ cut-in (testing both linear continuous and non-linear categorical parameterizations). Control cut-in speed and a site random effect were included in all models. For models with $\Delta$ cut-in speed as a categorical predictor, studies were binned into three discrete categories (1 = $\Delta$ cut-in values $\geq 0.5$ and $< 1.4$ m/s, 2 = $\Delta$ values $\geq 1.4$ and $< 2.6$ m/s, and 3 = $\Delta$ values $\geq 2.6$ m/s). Model selection was used to determine the optimal bin size (S2 Appendix).

For all models, the random effects meta-analysis calculates individual study weight using the variance-covariance matrix defined above, and the difference between the study mean effect and the grand mean effect. Among-site variance ($\sigma^2$) was estimated using a maximum likelihood approach [29]. Geographic region (Northeast, East, Midwest/West) was based upon EPA Level 2 ecoregions (https://www.epa.gov/eco-research/ecoregions) but consolidated to ensure enough studies per category for inclusion in models (Table 1). The northeast included Atlantic Highlands and Mixed Wood Plains ecoregions, the East included Ozark/Ouachita-Appalachian Forest ecoregion, and the Midwest/West included the Central Plains, Temperate Prairies, and West-central Semiarid Prairies ecoregions. Hub height was considered for inclusion as a covariate but had little variation across studies (n = 30 studies with hub height of 80 m). There were not enough data to consider interactions among these covariates. We examined between-study heterogeneity and model goodness of fit using Cochran's Q (QE), $\sigma^2$, and $I^2$ [30]. Model selection was performed based on Akaike's Information Criteria corrected for small sample size ($AIC_c$) values averaged across imputations, and model weights were calculated based on these values.

## Meta-analysis power analysis

To determine the number of studies required in a random effects meta-analysis to detect relative changes in RR reliably with $\Delta$ cut-in speed, we implemented a power analysis using a simulation approach [31]. We conducted meta-analysis power analyses for the non-linear categorical and linear continuous descriptions of the relationship between $\Delta$ cut-in speed and RR. For the categorical relationship power analysis, simulations were designed using the defined $\Delta$ cut-in speed categories to replicate the meta-analysis under multiple scenarios. The number of studies per $\Delta$ cut-in speed category (in aggregate, the total number of studies), fatality reduction parameter for the first $\Delta$ cut-in speed category ($\beta_0$), and the parameters representing the difference between the first category and the second and third categories ($\beta_1$, $\beta_2$), were varied across simulations. The following linear regression equation was used for the categorical model:

$$RR = \beta_0 + \beta_1 X_1 + \beta_2 X_2 + \varepsilon + \tau$$

where $X_1$ and $X_2$ are dummy covariates that represent $\Delta$ cut-in Categories 2 and 3, respectively. The uncertainty from the Gaussian error term ($\varepsilon$) and inter-study differences ($\tau$) were added by using a normal distribution with a mean of 0 and a standard deviation equal to that

observed in the fatality ratio of control-treatment data (Table 1; SD = 0.24). Study standard errors were used for weighting and drawn from a gamma distribution with the mean and variance of the observed standard errors in the database. For simulations using the linear continuous models, the same category framework was used to assign a Δ cut-in speed to each study randomly. Studies were randomly assigned a Δ cut-in speed from the same category they belonged to, drawn from the observed Δ cut-in speed values included in the meta-analysis. These values were then scaled (centered on zero) and used to build a linear model:

$$RR = \beta_0 + \beta_1 X_1 + \varepsilon + \tau$$

where $X_1$ is the scaled continuous Δ cut-in speed value for each study. A gamma distribution parameterized using the mean and variance of the standard errors was used to simulate standard errors for each study. This model is simpler than the methods we employed for the meta-analysis, as it assumes studies are independent and there are no site random effects. Simulating site-dependencies as sample size increased based on current knowledge was infeasible.

Four scenarios were simulated for each of the two model types with 5, 10, 20, and 30 studies per Δ cut-in category (32 total combinations simulated 10,000 times each; Table 2). The scenarios were selected *a priori* to explore our power to detect different relationships between fatality ratio and Δ cut-in. We included three scenarios thought to represent plausible hypotheses based on observed results to date: 1) a 25% linear decrease in fatality per 1 m/s increase in cut-in speed; 2) a 50% initial decrease in fatality with Category 1 Δ cut-in speed and subsequently stable fatality rates; and 3) an initial 50% decrease in fatality with Category 1 Δ cut-in speed and then 10% subsequent declines in fatality for Categories 2–3. The fourth scenario, a more extreme 50% exponential decrease per 1 m/s increase in cut-in speed, was intended to provide context for interpreting the results of other scenarios.

The statistical power of each parameter ($\beta_0$, $\beta_1$, and $\beta_2$) and sign error (the probability that the estimate was the same sign as the given parameter; [32]) were calculated to determine the effectiveness of the model in estimating the scenario parameters. Power was determined by examining whether the results were significantly different from the value of no effect (1 for $\beta_0$, and 0 for $\beta_1$ and $\beta_2$; $\alpha$ = 0.05), and sign error was computed by comparing the signs of the true parameter value and the estimated value.

Thus, this power analysis was designed to understand the efficacy of the current meta-analysis, above; estimate the number of studies needed to reduce uncertainty in the meta-analysis, and inform the likelihood that the *a priori* scenarios could generate the observed data.

**Table 2. Parameters used in simulation scenarios for the meta-analysis power analysis.**

| Model | Scenario | $\beta_0$ | $\beta_1$ | $\beta_2$ | *n* Studies |
|---|---|---|---|---|---|
| Non-Linear/categorical | 25% Linear Decrease | -0.29 | -0.41 | -1.1 | 5, 10, 20, 30 |
| | 50% Decrease then Stable | -0.69 | 0 | 0 | 5, 10, 20, 30 |
| | 50% Decrease then 10% Decline | -0.69 | -0.22 | -0.51 | 5, 10, 20, 30 |
| | 50% Exponential Decrease | -0.69 | -0.69 | -0.99 | 5, 10, 20, 30 |
| Linear/continuous | 25% Linear Decrease | -0.59 | -0.59 | | 5, 10, 20, 30 |
| | 50% Decrease then Stable | -0.52 | -0.27 | | 5, 10, 20, 30 |
| | 50% Decrease then 10% Decline | -0.7 | -0.5 | | 5, 10, 20, 30 |
| | 50% Exponential Decrease | -1.04 | -0.89 | | 5, 10, 20, 30 |

A total of four scenarios for each model type were run with different $\beta_0$, $\beta_1$, and $\beta_2$, values (the parameter estimates that define the relationship between fatality reduction and changes in cut-in speed) and differing numbers of studies per category (*n* Studies). Parameters are log-transformed.

## Fatality estimation power analysis

To understand the traits of effective curtailment experiments and provide guidelines for future studies, we used a hierarchical data simulation approach [33]. Three parameters were needed for this approach: fatality rates, carcass persistence rates, and detection probabilities. As the process for estimating these parameters is different for each, we simulated three data streams. First, the true number of bat mortalities was simulated using a Poisson process. New fatalities were generated each night for each turbine using a fatality rate per turbine-night as a Poisson mean. Second, carcass persistence rate was estimated using a carcass persistence trial format. Here, we used the exponential distribution to simulate the survival rates of 50 carcasses at the site based on the predefined median number of days of carcass persistence. Carcass searches were assumed to occur every three days for the duration of the study, and the survival probability of the carcasses was used to estimate daily carcass persistence probabilities for the survey. Third, searcher efficiency data were simulated based on 100 detection trials using a binomial distribution. These datasets were combined to determine the number of carcasses detected by the surveyors at each survey interval. Detection probability for each carcass was a function of carcass persistence, which changed in a time-dependent manner following the fatality event, and searcher efficiency, which was constant across time. The number of observed mortalities was determined using a binomial draw from the combined probability of persistence and detection for each fatality.

Forty-eight scenarios were used in this power analysis to explore the effects of effect size, study period, and carcass persistence on study design and were based upon information from AWWIC database curtailment studies (AWWIC studies in Table 1; accessed June 2019). Means and ranges of fatality rates, carcasses persistence estimates, number of survey nights, and survey turbines were defined based on data from previous studies (e.g., Table 1). Simulation parameters were selected based on averages and ranges from AWWIC. The curtailment treatment effect was defined as either a 25% or 50% reduction in fatality rates (n = 24 scenarios for each effect). These values were selected as a 50% reduction approximated the arithmetic mean fatality reduction of previous studies. The number of turbines (10 or 15), number of experiment nights (45 or 90), fatality rate (0.1 or 0.3 mortalities/turbine-night), and carcass persistence rate (3, 6, or 9 mean days of persistence) were selected based on values in studies included in the meta-analysis and were varied to determine the effect of these variables on study design power. The number of turbines and experiment nights were combined as turbine-nights to describe study effort. Carcass persistence values tended to be on the lower end of the range of observed values to test the power of these studies in more challenging environmental conditions. Detection probability was fixed at 50% for all studies, the approximate median of the described studies.

Data were simulated for each scenario using base functions in R v. 3.6 [15] and package *simsurv* [34]. Package *GenEst* [35] was then used to estimate each treatment group's true number of fatalities with the simulated datasets. This process was repeated 50,000 times to obtain consistent estimates of statistical power. This generalized fatality estimator ('GenEst') differs from those used by studies in the meta-analysis but is considered the current best practice for estimating fatality from wind turbines when the sample size is sufficiently large to estimate known biases [35]. The Bayesian posterior distribution of the number of fatalities for each treatment group was estimated using the function 'estM' in package *GenEst*. Simulated carcass observations, carcass persistence trial data, and searcher efficiency data were used as inputs along with assumed static values for the proportion of area searched (50% for all turbines) and the search schedule (once every three days for all turbines). The mean number of mortalities in the 25% and 50% reduction treatment groups (along with 95% credible intervals) were estimated using

parametric bootstrapping (n = 1000). The 95% credible interval of the difference of the *Gen-Est*-derived fatality estimates between these two groups was calculated to determine overlap with zero and used to estimate statistical power for each scenario, and was determined by subtracting the bootstrapped simulations for each treatment group. If a simulation study group did not detect any carcasses, we did not include it in the power analysis calculation.

## Results

Fatality ratios in studies included in the *meta-analysis* ranged from 0.04 (96% decrease in fatalities) to 1.00 (0% decrease in fatalities) with an arithmetic mean of 0.46 (53% decrease; n = 36 studies). When examining fatality ratios by Δ cut-in category (Fig 2), the mean fatality ratio for Category 1 (< 1.5 Δ cut-in speed) was 0.60 (40% decrease in fatalities, n = 12), Category 2 (≥ 1.5 and < 2.3 Δ cut-in speed) was 0.41 (59% decrease in fatalities, n = 18), and Category 3 (≥ 2.3 Δ cut-in speed) was 0.37 (63% decrease in fatalities, n = 6).

### Meta-analysis

Thirty-six studies (from 17 project sites) were included in the meta-analysis. The estimated fatality ratio across all studies (i.e., the estimate before controlling for Δ cut-in speed) was 0.37 (95% CI: 0.30–0.46, z = -6.63, p < 0.001; Fig 3), or a 63% reduction in fatalities in treatment

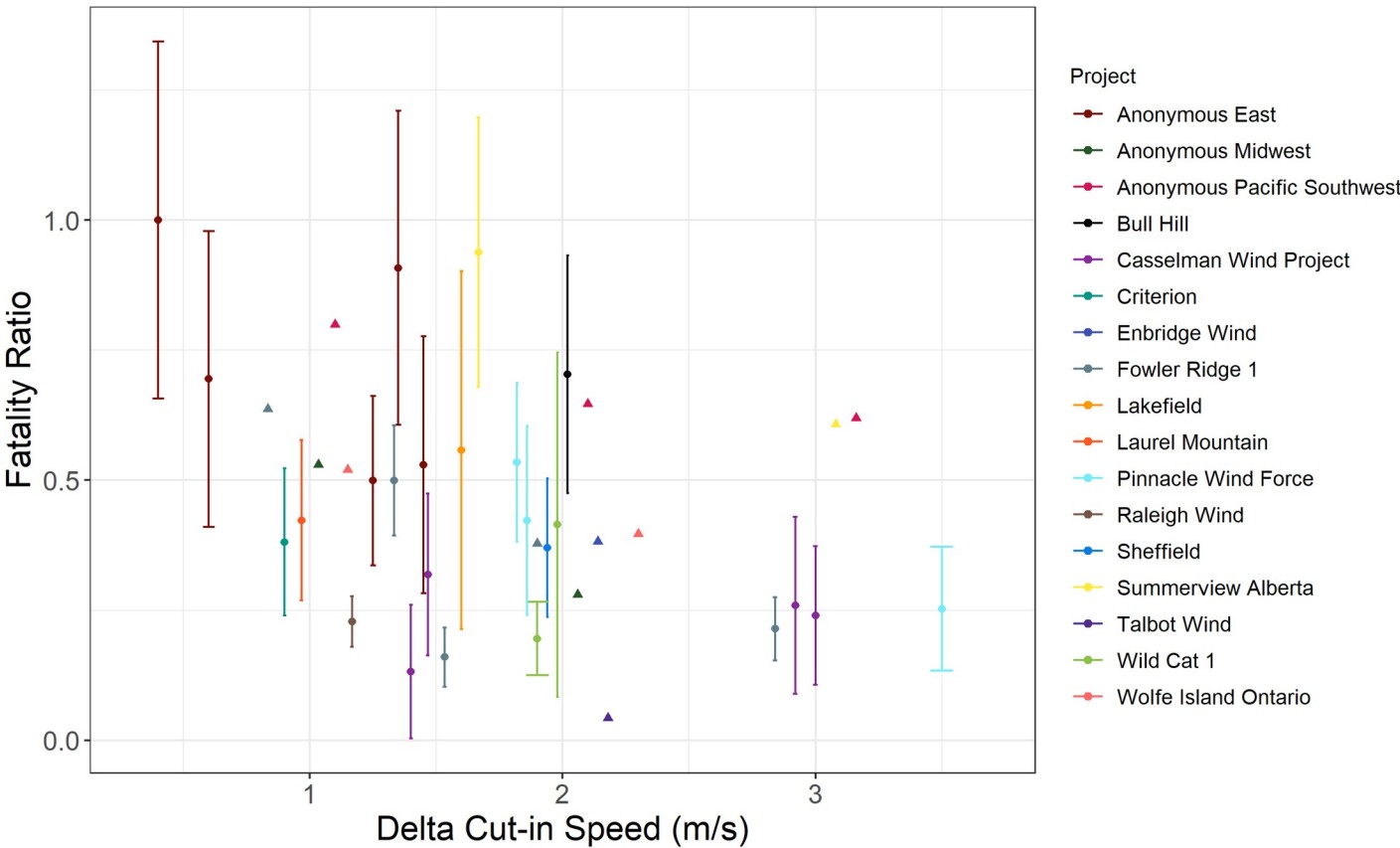

**Fig 2. The relationship between bat fatality ratio and Δ cut-in speed.** Using 36 studies conducted at 17 wind farm project sites in North America, a biplot describes the relationship between fatality ratio and Δ cut-in speed (calculated as a change in m/s between the treatment and control groups). Some project sites have multiple data points, as there were multiple years of experiments or multiple treatments tested within a year. Error bars represent the standard error of the fatality ratio for studies with estimated uncertainty (circles). Some studies (n = 12) did not provide a measure of uncertainty (triangles).

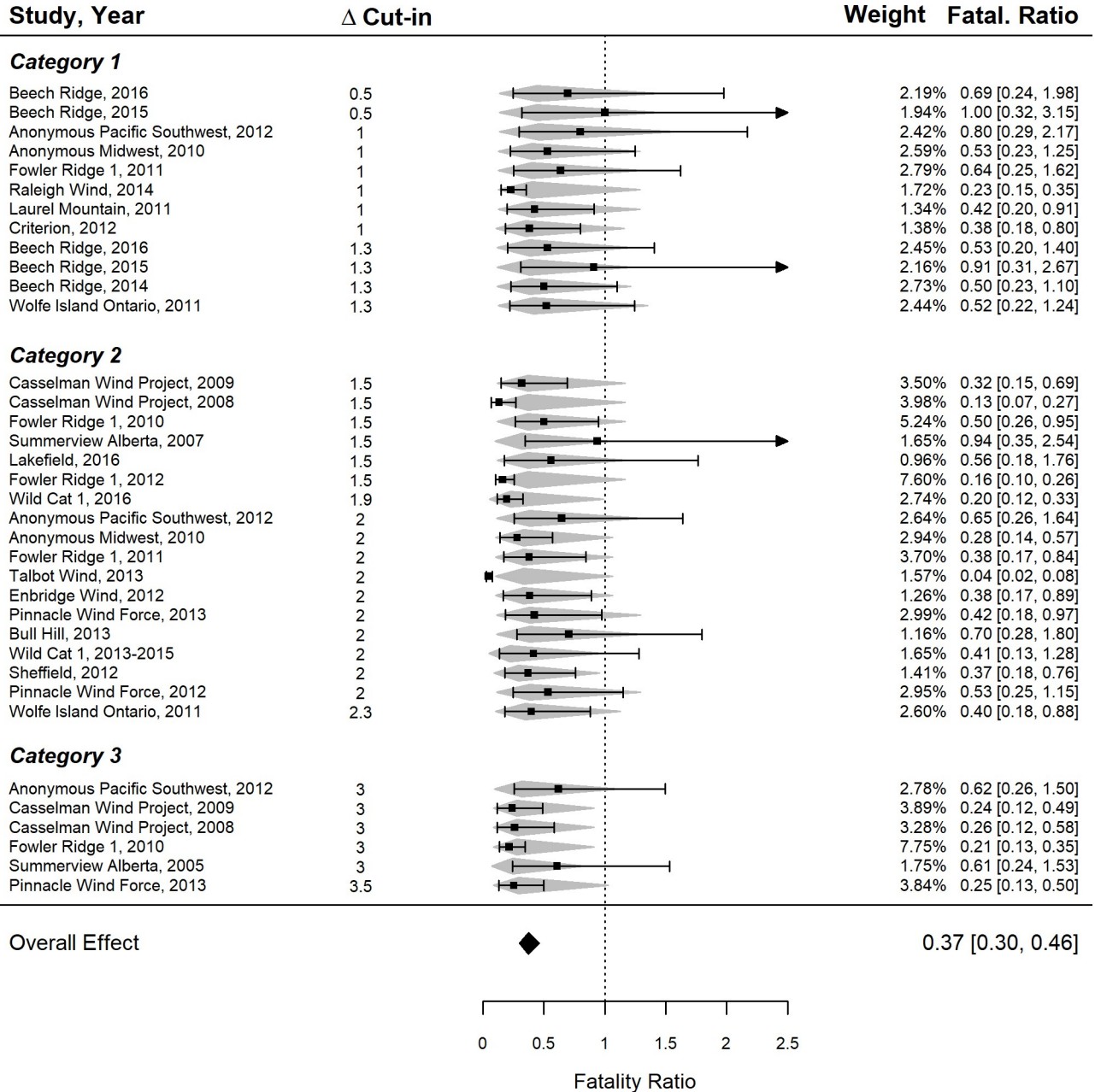

**Fig 3. The effect of curtailment regime on bat fatalities at terrestrial wind farms in North America from a random effects meta-analysis.** The plot shows the fatality ratio (black square) and 95% CI (error bars; the arrow indicates when the line extended off the scale) of individual studies along with the prediction interval for each study (grey diamonds). The 95% confidence interval of the overall effect is shown at the bottom (black diamond). Individual studies were weighted (out of 100%) based on study uncertainty (CI, in brackets). A fatality ratio of 1 indicates no difference in fatality rate between the control and experimental curtailment treatments.

groups as compared to controls. Models including Δ cut-in speed (continuous or categorical), control cut-in speed, and the random effect of site were included in the top models based on AIC$_c$ (Table 3). The linear model represented the best fit (AIC$_c$ = 73.27), while the categorical model had a higher likelihood but was ranked lower to additional complexity (AIC$_c$ = 74.04). However, both represented fits close to the base model with only control cut-in speed and the random effect (AIC$_c$ = 74.09). A funnel plot was used to discard concerns of publication bias

**Table 3. Model selection results for meta-analyses of the relationship between Δ cut-in speed and bat fatalities at terrestrial wind farms in North America.**

| Model | -logLik | $AIC_c$ | Weight | $I^2$ |
|---|---|---|---|---|
| Δ Cut-in (cont) + Control cut-in + 1\|Site | -31.99 | 73.27 | 0.30 | 68.63 |
| Δ Cut-in (cat) + Control cut-in + 1\|Site | -31.02 | 74.04 | 0.20 | 68.18 |
| Control cut-in + 1\|Site | -33.67 | 74.09 | 0.20 | 69.15 |
| Δ Cut-in (cont) + RD + Control cut-in + 1\|Site | -31.74 | 75.48 | 0.10 | 67.72 |
| Δ Cut-in (cont) + Region + Control cut-in + 1\|Site | -30.70 | 76.30 | 0.07 | 63.32 |
| Δ Cut-in (cat) + RD + Control cut-in + 1\|Site | -30.86 | 76.62 | 0.06 | 67.64 |
| Δ Cut-in (cat) + Region + Control cut-in + 1\|Site | -29.45 | 76.90 | 0.05 | 61.73 |
| Δ Cut-in (cont) + Region + RD + Control cut-in + 1\|Site | -30.54 | 79.08 | 0.02 | 62.42 |
| Δ Cut-in (cat) + Region + RD + Control cut-in + 1\|Site | -29.35 | 80.03 | 0.01 | 61.18 |

Models include Δ cut-in speed (m/s) as either a continuous (cont) or categorical (cat) variable. Best fit models were determined based on Akaike's Information Criteria corrected for small sample size ($AIC_c$), with–log-likelihood (-logLik), model weight, and percent of overall variation across sites due to heterogeneity ($I^2$) also shown.

and indicated that the Talbot Wind study exhibited high residuals. Upon further examination, Cook's Distance ($D_i$) for this study ranged in value (0.34–3.62) depending on the model but was generally low in the top continuous ($D_i = 0.41$) and categorical ($D_i = 0.90$) models. Removing this study did not change the top continuous or categorical model selection, though it did alter the significance of Δ cut-in speed in these models. Full dataset results are reported, recognizing the potential influence of this study (see Table S2-2 in S2 Appendix, for full comparison).

The top continuous model had a large and significant amount of residual heterogeneity among studies ($QE_{33} = 107.8$, p = <0.001), with an among-site variance estimate ($\sigma^2$) of 0.33, while the percentage of overall variation across sites due to heterogeneity ($I^2$) was 68.6%. Based on the top continuous model, the RR tended to decrease with increasing Δ cut-in (slope parameter β = -0.14, 95% CI: -0.29–0.01; Fig 4); this relationship nearly met the requirement for statistical significance (z = -1.80, p = 0.08). Control cut-in speed was not a significant covariate (β = -0.13, CI: -0.43–0.17, z = -0.84, p = 0.41). The categorical response to Δ cut-in speed also had a significant amount of residual heterogeneity between studies ($QE_{32} = 102.0$, p < 0.001), with an among-site variance estimate ($\sigma^2$) of 0.32, while the percentage of overall variation across studies due to heterogeneity ($I^2$) was 68.2%. When examining fatality reduction by Δ cut-in speed, the model predicted a fatality ratio estimate for Category 1 of 0.48 and represented a significant reduction in fatality rates (β = -0.75, CI: -0.1.05 - -0.46, z = -3.48, p = 0.001). The model estimates for fatality ratios for Categories 2 and 3 were 0.30 and 0.28 respectively, but the marginal change of increasing Δ cut-in from Category 1 to Category 2 ($\beta_1$ = -0.43, CI: -0.85–0.01, z = -1.98, p = 0. 06) was marginally nonsignificant, while the change from Category 1 to Category 3 was significant ($\beta_2$ = -0.55, CI: -1.06- -0.05, z = -2.17, p = 0.04; Fig 4). Control cut-in speed was not a significant covariate (β = -0.11, CI: -0.41–0.19, z = -01.73, p = 0.47). The base model had no significant covariates but it had more residual variation among studies (QE = 112.6, p < 0.001) though $I^2$ and $\sigma^2$ were similar to the others at 69.2% and 0.33, respectively.

The top models via model selection did not include rotor diameter nor geographic region, and these covariates were not significant in the global continuous model (rotor diameter: CI: -0.39–0.22, z = -0.57, p = 0.57; geographic region: Midwest vs. East, CI: -0.48–1.00, z = 0.70, p = 0.49; Northeast vs. East, CI: -1.06–0.41, z = -0.86, p = 0.40) or global categorical model (rotor diameter: CI: -0.37–0.23, z = -0.45, p = 0.66; geographic region: Midwest vs. East, CI: -0.36–1.13, z = 1.01, p = 0.32, Northeast vs. East, CI: -1.01–0.46, z = -0.73, p = 0.46).

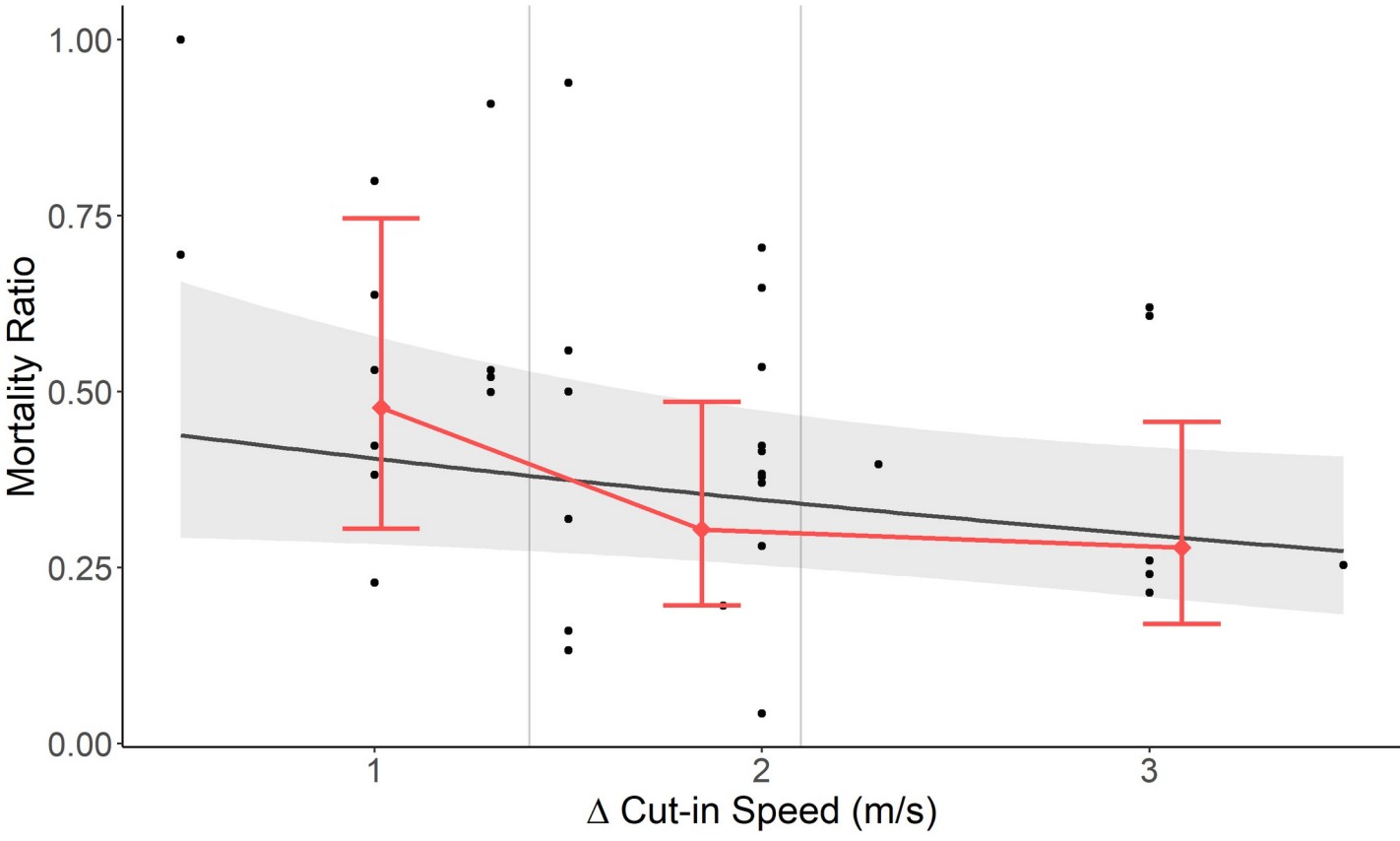

**Fig 4. The estimated effects of Δ cut-in speed on bat fatality ratio at North American wind energy projects.** The meta-analysis estimated both continuous (black line) and categorical (pink) effects of Δ cut-in speed on bat fatality ratio from the top two models. Black dots represent fatality ratios for individual studies; note that uncertainty in individual study estimates, which influenced the meta-analysis parameter estimates, are not shown here (see Table 1 for these values). Categorical model points are based on mean Δ cut-in speed for the category. Error bars are 95% confidence intervals of estimates. Gray vertical lines represent Δ cut-in categories.

## Meta-analysis power analysis

Power analysis of the categorical model revealed that for most scenarios, five studies were required to have adequate statistical power ($>0.8$) to determine an effect of curtailment on the fatality ratios for Category 1 ($\beta_0$; 0.5–1.3 m/s Δ cut-in; Fig 5). The exception was the 25% linear decrease scenario, which required over 30 studies to achieve adequate power due to smaller changes at lower Δ cut-in speeds. The statistical power of $\beta_1$ and $\beta_2$ (Δ cut-in Categories 2–3) were more variable across scenarios (Fig 5). For $\beta_1$ (1.5–2.3 m/s Δ cut-in speed), only the 50% exponential decrease achieved sufficient power, and it required 20 studies per category (60 studies overall). For $\beta_2$ (3–3.5 m/s Δ cut-in speed), the 50% exponential decrease and 25% linear scenarios had sufficient power at 10 studies per category (30 studies overall), and the 50% initial decrease with 10% decline scenario had sufficient power at 30 studies per category (90 studies overall). Sign error decreased with increasing sample size for all parameters except those set at zero ($\beta_1$ and $\beta2$ in the 50% decrease then stable scenario) and decreased below 10% at 10 studies per category for most other parameter estimates.

In comparison, the continuous model had higher power for assessing changes, particularly for constant or increasing relationships between RR and Δ cut-in (Fig 6). Power to detect linear trends ($\beta_1$), particularly for the scenarios with decreases at Δ cut-in speeds greater than 1.3 m/s, was greater than 0.8 even with only five studies per category. Only the 50% then stable

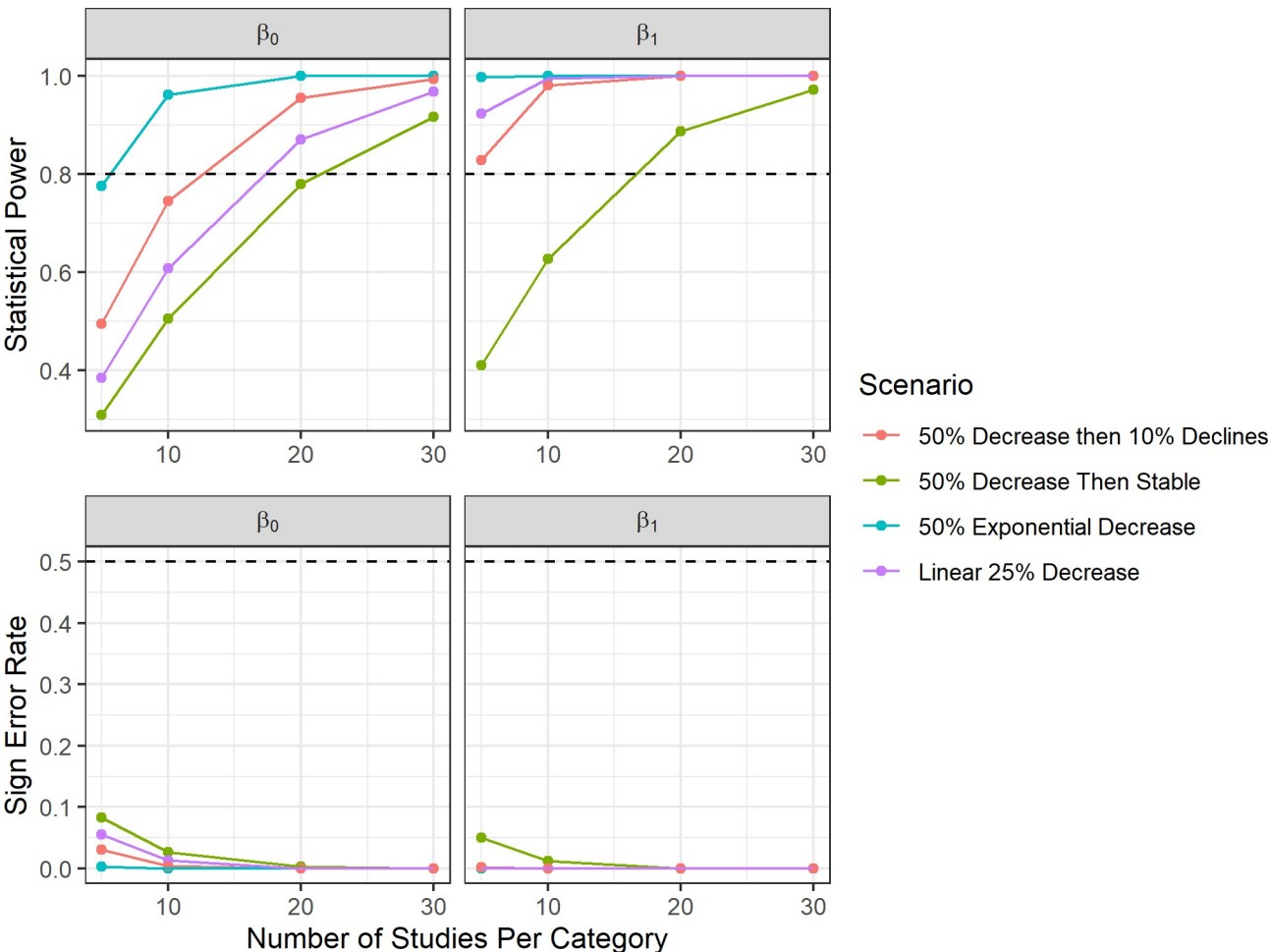

**Fig 5. The relationship of statistical power and sign error with sample size in the categorical meta-analysis power analysis of curtailment studies.**
We examined the relationship between the number of studies per category of Δ cut-in speed (Category 1 = $\beta_0$ = 0.5–1.3 m/s Δ cut-in speed, Category 2 = $\beta_1$ = 1.5–2.3 Δ cut-in speed, Category 3 = $\beta_2$ = 3–3.5 m/s Δ cut-in speed) and 1) the statistical power to detect change between categories (top), and 2) the rate at which models would be expected to incorrectly predict the sign of parameter estimates (bottom). Colors represent different curtailment regime scenarios. The horizontal dashed lines represent the 0.8 power threshold and 50% sign error threshold, respectively.

scenario showed poor power, likely needing 20–30 studies per category to measure the decrease accurately. The average value (or intercept, $\beta_0$) was more difficult to precisely estimate, though this parameter is less important to the present study as it does not estimate the change in effect with Δ cut-in. Sign error was low across all scenarios; it was lower than 10% whenever the number of studies per category was greater than 10.

## Fatality estimation study power analysis

For individual curtailment experiments at wind facilities, many factors influenced these studies' statistical power and sign error. More turbine-nights increased the power of studies in all scenarios (Fig 7). However, the importance of turbine-nights varied with several variables outside of researcher control, such as effect size and carcass persistence. With a 25% fatality reduction between experimental and control treatments (Fig 7A and 7C), no tested scenario achieved statistical power of 0.8 when the control fatality rate was low (0.1 mortalities/turbine-night; Fig 7A). For scenarios with a 25% reduction in fatalities, statistical power was high only

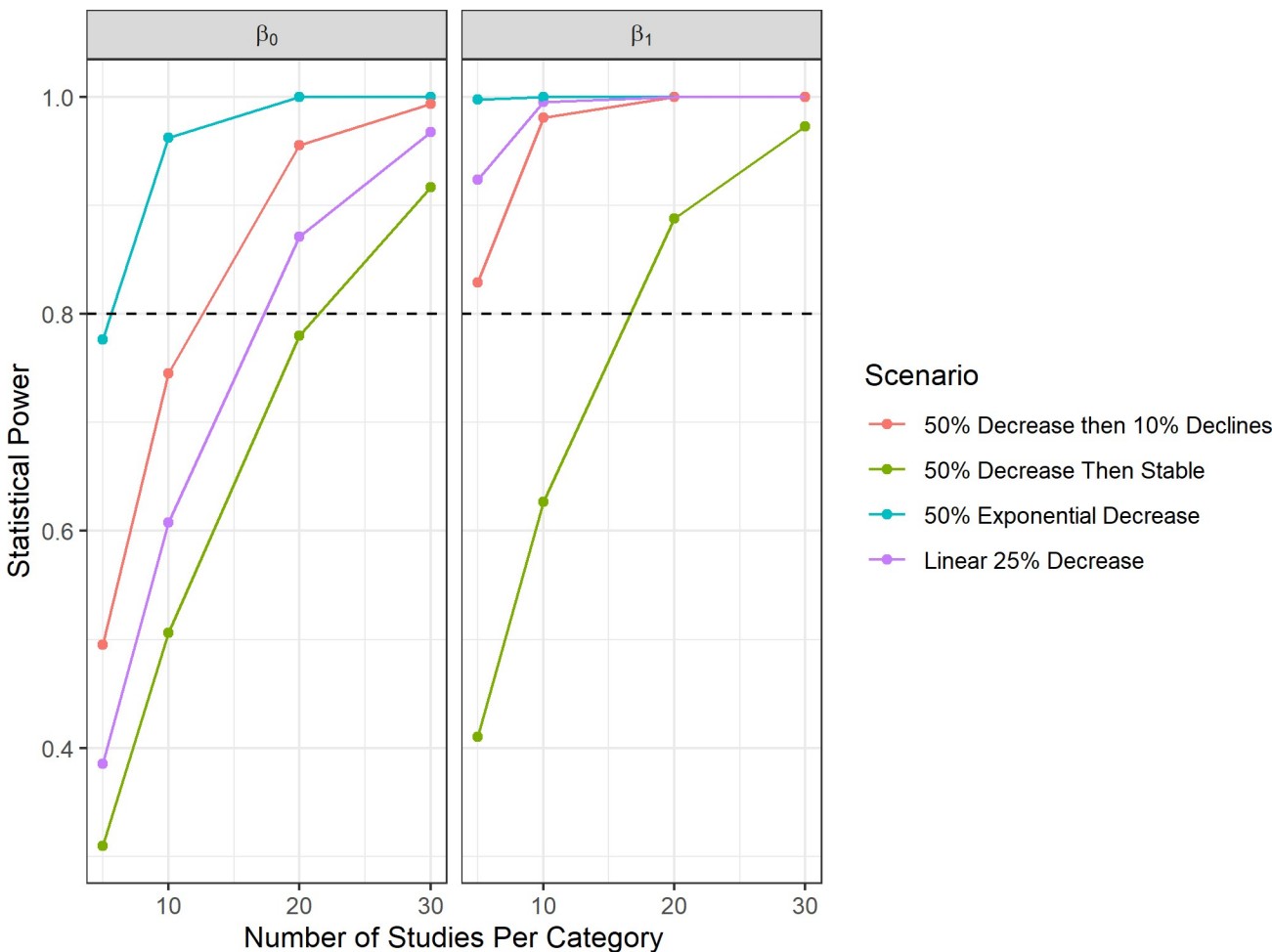

**Fig 6. The relationship of statistical power and sign error with sample size in the continuous meta-analysis power analysis of curtailment studies.**
We examined the relationship between the number of studies per $\Delta$ cut-in speed category (Category 1 = $\beta_0$ = 0.5–1.3 m/s $\Delta$ cut-in speed, Category 2 = $\beta_1$ = 1.5–2.3 $\Delta$ cut-in speed, Category 3 = $\beta_2$ = 3–3.5 m/s $\Delta$ cut-in speed) and 1) the statistical power to detect change in fatality ratio (top), and 2) the rate at which models would be expected to incorrectly predict the sign of parameter estimates (bottom). Colors represent different curtailment regime scenarios. The horizontal dashed lines represent the 0.8 power threshold and 50% sign error threshold, respectively.

when fatality rate, carcass persistence, and turbine-nights were also high. For studies in the 50% fatality reduction scenarios (Fig 7B and 7D), statistical power was more resilient to changes in sampling period and carcasses persistence than the lower-reduction scenarios. Statistical power was above the 0.8 threshold across almost all scenarios with high fatality rates (0.3 fatalities per turbine-night), and a large number of turbine-nights yielded strong statistical power even when the fatality rate was lower (Fig 7B). Sign error followed a similar pattern, with errors occurring more often when fatality rates and fatality reduction from curtailment were low (Fig 7C). When fatality reduction was 50%, sign error was almost always less than 10% (Fig 7D). In summary, these simulation results suggest that many curtailment study designs could effectively detect differences between treatments in scenarios with high fatality rates and high carcass persistence. None of the tested study designs effectively detected change when fatality reduction and carcass persistence were low. Based on the studies in our database (which had a median number of 14 turbines and 75 experimental nights, 1050 turbine-nights, and 16 of 36 studies with percent fatality reductions <50%), many studies could have low power and high sign error if fatality and carcass persistence rates were low.

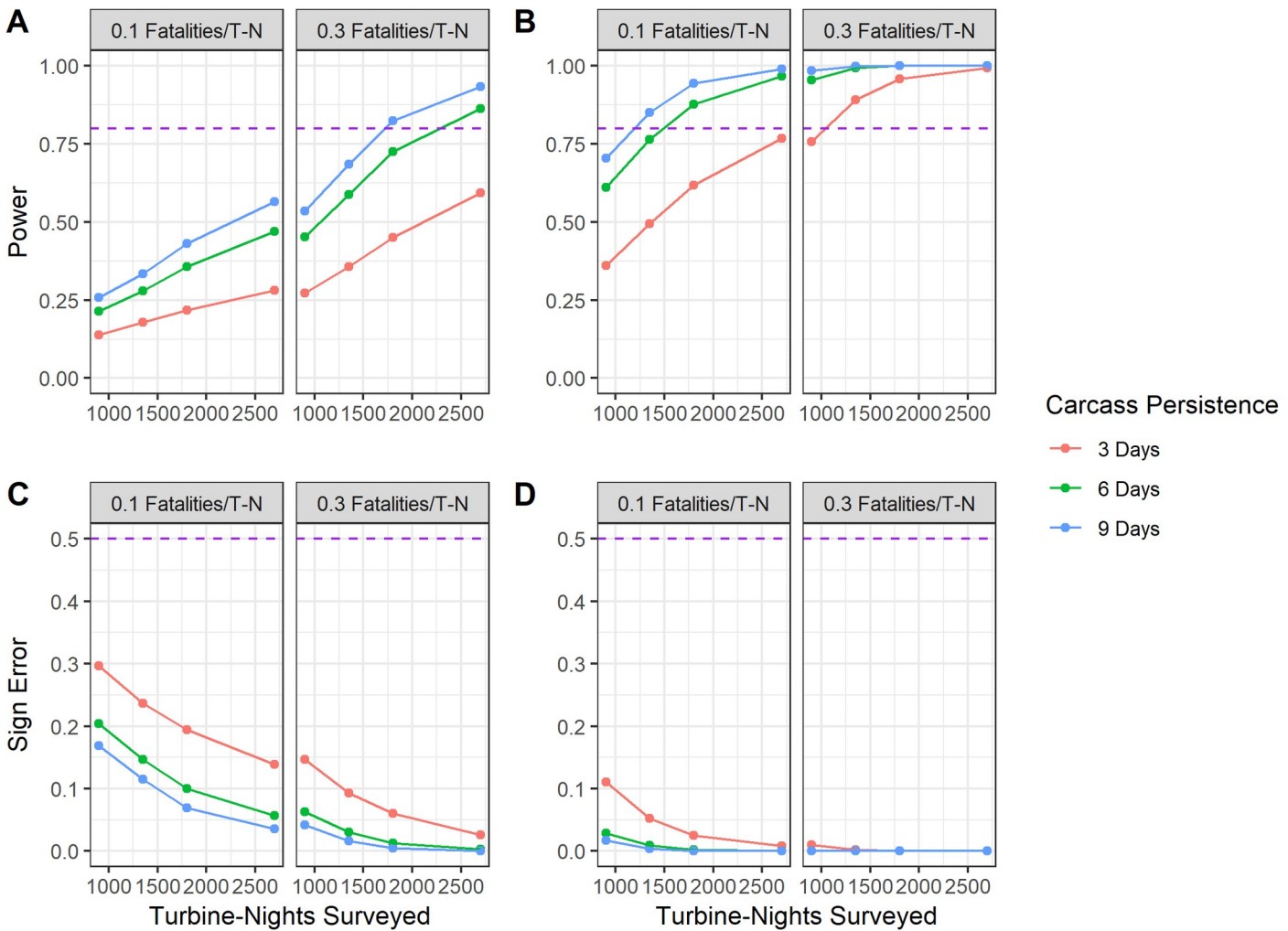

**Fig 7. The relationship of statistical power and sign error with sample size for individual curtailment experiment scenarios.** Using a simulation approach (n = 50,000) using the Generalized Mortality Estimator (*GenEst*), we found sample size (A, B) and sign error(C, D) were influenced by fatality rates, survey effort, and carcass persistence. Variation in power across turbine-nights of study (T-N), fatality rates, and carcass persistence (in mean days of persistence) is shown when curtailment is simulated to reduce fatality by 25% (A, C) and 50% (B, D). Simulations assume a three-day search interval for fatality searches and 50% searcher efficiency.

## Discussion

We found strong evidence that turbine curtailment reduces fatality rates of bats at wind farms that have implemented the technique, as supported in the previous review by Arnett et al. [36]. The relationship between the magnitude of treatment (e.g., Δ cut-in speed) and reduction in fatalities was more difficult to assess with the available dataset. The top three models all had a similar degree of AIC$_c$ support in the meta-analysis, and each included a different relationship between Δ cut-in speed and fatality ratio (linear, categorical, and none). The near-significance of Δ cut-in speed for decreasing RR in the linear model, and the statistically significant difference between Δ cut-in Category 1 (Δ cut-in speed of 0.5–1.3 m/s, fatality ratio 0.48) and Category 3 (Δ cut-in speed > 2.6 m/s, fatality ratio of 0.28) in the categorical model, collectively suggest that that increasing cut-in speed may provide further fatality reduction. Moreover, the latter model suggests a non-linear relationship between these variables. However, several issues complicate our inference. First, the model quantifying this effect was not the most likely in our

model selection framework. Second, the presence of a high-leverage datapoint (which altered some results when removed) suggests that these results should be treated with some caution. Third, the base model is also included in the top model set, representing the simplest choice. Thus, the effect of (Δ cut-in speed is large enough to meet a statistically significant threshold, but only when parsimony is ignored. Taken together, we interpret these results with caution and suggest that an effect of cut-in speed on bat fatalities is likely, but more information is needed.

Within the context of the *meta-analysis power analysis*, we generally had the statistical power to consistently detect reductions of ~50% per 1 m/s Δ cut-in speed with our current sample size. Our results are consistent with these estimates in that we were able to detect the effect of Δ cut-in speed when the magnitude of effect approached the 50% threshold. The lack of effect in the continuous linear model suggests that the 25% linear decrease, 50% decrease then 10% further decline, and 50% exponential decrease scenarios all appear unlikely based on the meta-analysis results and that the 50% decline then stable scenario may be the closest tested scenario to the true relationship between these variables.

Other concurrent efforts to examine the efficacy of operational curtailment have also found that increased curtailment speeds significantly affect bat mortality [37]. Whitby et al. [37]'s recent meta-analysis produced an estimated bat fatality reduction between 33–79%, consistent with our estimate of 0.37, or a 63% decrease in bat fatalities (95% CI 0.30–0.46, or 54–70%). They suggested a possible linear relationship between curtailment and fatality reduction, estimating a 33% reduction in fatalities for every 1 m/s increase in cut-in speed. This difference from our findings may in part be due to the use of different methods, but it could also be due to differing datasets. Our meta-analysis average was closer to the upper confidence interval of their estimate; this implies that our studies were more effective on average and perhaps had a lower marginal value for increasing cut-in speed.

Uncertainty in fatality reduction estimates was variable across studies. While this imprecision was accounted for in the meta-analysis framework and propagated into parameter estimates, uncertainty should be minimized through careful study design to maximize the value of each study in future meta-analyses. Through our *fatality estimation power analysis*, we found that with high fatality rates (≥0.3 fatalities per turbine-night) and carcass persistence (≥6–9 days), experimental studies were consistently successful in detecting 25–50% fatality reductions. However, studies with low fatality rates (0.1 fatalities per turbine-night) and carcass persistence (3 days) were not adequate to detect 25% differences in fatality rates between treatment and controls groups even with high numbers (>2500) of turbine-nights. These results suggest that while effective monitoring studies can be conducted when assumptions are met (e.g., detection probability is at least 50%), some studies could have low statistical power when using the *GenEst* modeling framework. Continuing to conduct high-quality curtailment experiments with a large number of experimental turbine-nights, particularly if fatality rates are expected to be low, would better estimate the effect of this operational mitigation approach and inform conservation and management activities for bats [38].

## Assessing the role of cut-in speed on bat fatalities

While the overall effect of curtailment on bat fatality was clear, the relative effect of incrementally larger increases in curtailment cut-in speed was not. The results of the meta-analysis did not provide clear evidence as to whether Δ cut-in speed affected fatalities reduction in a linear or non-linear manner. The non-linear categorical model was the only model with statistically significant parameter estimates, but the linear continuous model was selected as the best-fit model via $AIC_c$ due to parsimony. Moreover, while the non-linear model does provide evidence that cut-in speed is important, it does not definitively suggest that the relationship

between Δ cut-in speed is non-linear. While a non-linear relationship seems likely given the current estimates of differences in the Δ cut-in speed categories, confidence intervals are overlapping across some categories, and we have the most certainty in the difference between Δ cut-in speed categories 1 and 3.

A likely source of this uncertainty is a lack of studies available, given the size of the effect. At the current sample size (n = 36), the meta-analysis power analysis does confirm that we had enough power to detect large differences in fatality reductions, but not smaller effects. We have the power to detect differences in the categorical framework at Δ 2 m/s for the scenarios with the highest magnitude decrease (25% linear and 50% exponential) and to detect linear trends for these same scenarios as well as the 50% initial decline/10% long-term decline scenario. Thus, the marginal outcome of the linear model either indicates that these scenarios are not plausible or the non-significant result was unlikely. However, the *meta-analysis power analysis* used a simpler model than the *meta-analysis*, underestimating the site-dependencies in the actual data and likely overestimating the power of the current analysis. Given that the power of many scenarios significantly exceeded the 0.8 threshold, we do not anticipate this greatly affecting these results, but caution should be used in interpreting marginal cases.

We found no evidence that factors other than Δ cut-in speed affect fatality reductions. Though included in all final models, control cut-in speed was not an important predictor, signifying that the RR approach was effective in standardizing effect sizes across studies. Neither rotor diameter nor geographic region explained significant variation in RR, which may relate to the scale of the variable; in the case of geographic region, for example, we used broad geographic areas rather than ecoregions due to sample size limitations. Previous research has also indicated that bat fatalities increase exponentially with tower height [8, 39], but we were unable to include hub height due to a lack of variation in the dataset. Bat mortality risk has also previously been related to habitat characteristics such as forested areas, slope, temperature, and humidity [40, 41], and mountain ridges have been recognized as important during migration [42]. These types of fine-scale environmental data should be collected and provided for future syntheses, as more information on how these factors affect curtailment efficacy will be important for effective siting and mitigation of wind energy developments. Additionally, testing curtailment efficacy at locations with lower overall fatality rates could be instructive (though, as shown in the *fatality estimation study power analysis*, somewhat challenging to implement), as curtailment studies are often implemented at projects with fatality rates high enough to elicit conservation concern.

Differentiation of fatality rates by species or species group could also help reduce uncertainty in turbine siting. Species-level traits such as migratory strategy, dispersal distance, and habitat association likely play an important role in fatality risk [43]. For instance, long-distance migrants such as hoary bats, silver-haired bats, and eastern red bats comprise a majority of fatalities at terrestrial wind energy facilities in North America [3, 8]. Project-specific risk is thus likely correlated with species distributions, migratory routes, and flight heights, among other characteristics [44]. Incorporating species-level information could improve our understanding of bat fatality reduction, but this would require that species-level fatality estimates, or at least species-group fatality estimates (i.e., migratory tree bats vs. *Myotis* spp.), be reported from curtailment studies. The studies included in our analysis did not consistently report such estimates, often due to insufficient sample size.

## Recommendations for future studies

If the implementation of operational curtailment > 1.5 m/s above manufacturer specifications continues at wind facilities, additional experiments should be conducted to understand the relative benefit of these increased cut-in speeds for reducing bat fatalities. Estimates from the

*meta-analysis power analysis* suggest that 25–55 additional studies would be needed to effectively exclude the possibility of an additional 20% reduction in fatalities beyond what is seen at lower Δ cut-in speeds. Fewer studies might be needed if experiments are conducted at higher cut-in speeds or concurrently compare multiple treatment groups against a control. At the individual study level, statistical power is dependent on many factors outside of the control of study designers (e.g., fatality rates and carcass persistence). Prior knowledge of these parameters is valuable for designing effective studies, particularly if carcass persistence rates are expected to be lower than average (e.g., due to high scavenging activity at a site).

To facilitate inclusion of studies in future meta-analyses, curtailment experiments should report fatality estimates for both control and treatment groups, as well as carcass persistence rates, searcher efficiency, search frequency, search area coverage, number of turbine-nights of study, curtailment regime (including whether feathering occurred), and turbine makes/models, with associated uncertainty values when relevant. When sample size allows, fatality estimates should be reported by species or species group (e.g., *Myotis*) rather than for all bat species combined to facilitate taxon-specific assessments of curtailment efficacy.

Newer operational minimization strategies have been developed to achieve fatality reductions with lower energy loss than the type of curtailment examined here [45, 46]. "Smart" curtailment strategies, for example, use additional environmental data besides wind speed to inform curtailment implementation [47–49], and deterrent systems attempt to discourage bats from approaching turbines [11, 49–51]. The efficacy of such approaches needs further evaluation and comparison current approaches, they may eventually represent effective alternatives, or additions, to wind speed-based curtailment approaches.

## Conclusions

Given the scope of bat fatalities at terrestrial wind farms in North America [3, 52], we must learn more about the management effectiveness of curtailment, particularly at higher cut-in speeds. The results of our *meta-analysis* suggest that operational curtailment effectively reduces bat fatalities at terrestrial wind energy facilities and that higher cut-in speeds can reduce fatalities further, though we lack overwhelming evidence for the latter. The *meta-analysis power analysis* suggested that we could detect >50% changes in fatality rates with the current sample size, which gives us confidence that the marginal effect of increasing cut-in speed is below that threshold. However, even smaller changes in fatality rate could have cumulative implications for species of conservation concern, and an improved understanding of the incremental value of higher Δ cut-in speeds could help to inform future management decisions.

If curtailment continues to be a common strategy at wind speeds at ~5 m/s or above (i.e., Δ cut-in speed of >1.5 with a standard factory cut-in speed of 3.5 m/s), we would recommend conducting additional experimental curtailment studies with treatments at these higher cut-in speeds to strengthen our understanding of the relationship between increasing cut-in speeds and bat fatality rates. Such studies must be carefully designed, ideally using an adaptive management framework [53], to consider such variables as the expected fatality rate and carcass persistence rate when selecting a search interval and defining the number of turbine-nights to monitor. Studies at sites with expected low fatality rates and low carcass persistence, in particular, must be carefully designed, and power analyses are a necessary tool to ensure adequate statistical power to detect changes across treatment and control groups.

## Supporting information

**S1 Checklist.**
(DOCX)

**S1 Table. Downloadable version of Table 1 with additional covariates appended.**
(CSV)

**S1 Appendix. R code for analysis.**
(DOCX)

**S2 Appendix. Assessing meta-analysis sensitivity to category choice and outliers.**
(DOCX)

## Acknowledgments

We want to thank the American Wind Wildlife Information Center (AWWIC) database manager, Ryan Butryn, for data collation and management, and the AWWI project managers for logistical support and input on the draft report. Four anonymous reviewers provided valuable feedback on earlier drafts of this study. We would also like to thank all the data contributors who conducted curtailment studies and made the data available to this project.

## Author Contributions

**Conceptualization:** Evan M. Adams, Julia Gulka, Kathryn A. Williams.

**Formal analysis:** Evan M. Adams, Julia Gulka.

**Funding acquisition:** Evan M. Adams.

**Methodology:** Evan M. Adams, Julia Gulka.

**Project administration:** Kathryn A. Williams.

**Supervision:** Evan M. Adams, Kathryn A. Williams.

**Visualization:** Evan M. Adams, Julia Gulka.

**Writing – original draft:** Evan M. Adams, Julia Gulka, Kathryn A. Williams.

**Writing – review & editing:** Evan M. Adams, Julia Gulka, Kathryn A. Williams.

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
