## [Decision Letter · Decision Letter 0]

14 Sep 2021

PONE-D-21-22448Examining the effectiveness of blanket curtailment strategies in reducing bat fatalities at terrestrial wind farms in North AmericaPLOS ONE

Dear Dr. Adams,

Thank you for submitting your manuscript to PLOS ONE. After careful consideration, we feel that it has merit but does not fully meet PLOS ONE’s publication criteria as it currently stands. Therefore, we invite you to submit a revised version of the manuscript that addresses the points raised during the review process.

 Please submit your revised manuscript by Oct 29 2021 11:59PM. If you will need more time than this to complete your revisions, please reply to this message or contact the journal office at plosone@plos.org. Please include the following items when submitting your revised manuscript:A rebuttal letter that responds to each point raised by the academic editor and reviewer(s). You should upload this letter as a separate file labeled 'Response to Reviewers'.A marked-up copy of your manuscript that highlights changes made to the original version. You should upload this as a separate file labeled 'Revised Manuscript with Track Changes'.An unmarked version of your revised paper without tracked changes. You should upload this as a separate file labeled 'Manuscript'.

We look forward to receiving your revised manuscript.

Kind regards,

Ignasi Torre

Academic Editor

PLOS ONE

Journal Requirements:

The authors are grateful to the Wind Wildlife Research Fund for providing financial support to make this work possible. 

This study was funded by the AWWI Wind Wildlife Research Fund (Award B-03. https://awwi.org/wind-wildlife-research-fund/). All authors were funded by this award and the funders provided four anonymous reviewers for an earlier version of the manuscript.

Additional Editor Comments:

Our author guidelines for systematic reviews/meta-analyses are at http://journals.plos.org/plosone/s/submission-guidelines#loc-systematic-reviews-and-meta-analyses.

Reviewers' comments:

Reviewer's Responses to Questions

**Comments to the Author**

1. Is the manuscript technically sound, and do the data support the conclusions?

Reviewer #1: Yes

Reviewer #2: No

2. Has the statistical analysis been performed appropriately and rigorously? 

Reviewer #1: Yes

Reviewer #2: No

3. Have the authors made all data underlying the findings in their manuscript fully available?

Reviewer #1: Yes

Reviewer #2: No

4. Is the manuscript presented in an intelligible fashion and written in standard English?

Reviewer #1: Yes

Reviewer #2: Yes

5. Review Comments to the Author

Reviewer #1: The authors conducted a very impressive meta-analysis. The manuscript is extremely well written and clear, although the method section is very long to read and complex to understand. This study not only provides a concrete assessment of the effectiveness of wind turbines curtailement strategies to limit bat fatality risk, but also provides power analyses and simulations that provide extremely important insights into our ability to detect such effects as a function of parameters at the meta-analysis level but also at the study level. This study is in my opinion extremely rich, and of a very high scientific level.

I don't have many comments, I enjoyed reading this study although it is not easy to understand (I don't see how it could be otherwise with such rich results/analyses and such a complex theme).

Please find some specific comments in the pdf.

Reviewer #2: The study uses meta-analysis and a power analysis to attempt to estimate the efficacy of wind turbine curtailment to reduce bat fatalities. The topic is timely and important given that it has been widely reported that wind turbines kill large numbers of bats and numerous individual studies have been conducted to determine the reduction in bat fatalities by changing the cut-in speeds of turbines. However, there are some structural flaws to the way the analysis appears to be conducted and the writing is often quite opaque and confusing, making it difficult to carefully discern exactly what was done and the assumptions and choices made by the authors. The approach to the meta-analysis and the way those methods and results are presented and then interpreted raise serious concerns. I’ve made exhaustive comments and I would encourage the authors to rethink the categorical vs linear approach to the meta-analysis, as presenting both fails to lend additional insight and violates some core assumptions of how model selection works. The flaws and problems with the methods need to be addressed. In general, the analysis and interpretation of the results is confusing and appears forced and on shaky grounds inferentially.

Title + Line 9 - suggest removing the term 'blanket'. There's no additional meaning/value added by the word blanket in this context. I'm aware that is a jargon/term used by wind industry to try and distinguish from 'smart' curtailment but that distinction is arbitrary and I suspect designed to make it sound like curtailment strategies based on wind speeds are insufficiently sophisticated.

Line 9 - 'accepted' -- really? Accepted by whom? Certainly not universally accepted by the wind industry.

Line 11 - reduces impacts? or reduces fatalities?

Line 14-15 - What is meant by 'tested multiple statistical models' ? did you use statistical models to test hypotheses? Or use multiple statistical models to explore relationships? Testing models seems odd wording and makes the intent unclear.

Line 16-20 - seems a bit detailed for an abstract

Line 22 – response ratio is the way to measure the effect size, not an ‘approach’

Line 24 –the power analysis shows power is low if relationship is <50% (which would be the case with small increases in cut-in speed), so is this interpretation minimizing the existing evidence?

Line 33 –explanation is not the right word here. Turbine attraction is a hypothesis that has not been proven, but is what we think may be happening (without knowing the ‘why’). The explanation for fatalities is that blades hit and kill bats

Line 38 –specify conservation concern is for populations due to cumulative impacts

Line 39 - not sure what is meant by 'accepted' ; also the citation here doesn’t seem to relate to the sentence as written re operational minimization as ‘accepted’ tactic?

Line 42 – reduce blade spinning rates below cut-in speed. The way this is written reads like blade spinning is reduced all the time vs below specified cut-in speed.

Line 43- replace ‘still’ with ‘can’ spin

Line 43/44 – make feathering a new sentence and define it more clearly for clarity

Line 48 – this sentence is misleading and incorrect as written. The AWWI summary report cited does not report fatalities associated with turbines operating under curtailment regimes (see Page 9 of AWWI summary report, criteria 2: Turbines operated at normal procedures (e.g., studies conducted while turbines were operating under a curtailment regime were not included)). The statement “a great deal of variability has been reported in the level of fatality reduction achieved by curtailment” needs clearer attribution and substantiation with data/appropriate reference.

Line 49 - replace 'impacts' with 'fatalities'

Line 49 - the wording here is verbatim as in the abstract .. what does 'exact nature of the relationship' mean? Seems like this is written to justify the work but I'm not sure the point of a meta-analysis is to get at the 'exact nature' ... isn't it more to determine general patterns or estimate average/mean effects from site-specific studies?

Line 51 - the fact that you had to put 'blanket' in quotes supports my earlier comment to just delete the term. It's jargon without much useful purpose. What does the leading preposition phrase, "For this study..." mean here in this context? Doesn't curtailment have both operational and financial implications for wind facility operators outside the context of this study? The cited study does not use the word “blanket”

Line 53 - the phrase 'exact nature' stands out to me here as well.. it's very rare we can estimate or determine the exact nature of anything.. almost all relationships have some uncertainty associated with them. This sentence could be simplified by reducing to, "The trade-offs between turbine energy productions and bat fatality minimization are poorly understood."

Line 55 - the word "Still" doesn't work here. Delete. "this type of assessment" - isn't clear what type of assessment you are referring to.

Line 57 - delete 'blanket' (here and elsewhere used in the manuscript)

Line 57-59 – this confuses the reasons for raising cut-in speed to 6.9m/s. The reality is that curtailment, especially high wind speed curtailment, is done as an avoidance measure to prevent fatalities of endangered species. 6.9 m/s is not often implemented as a minimization method for non-listed species.

Line 62 – meta-analysis is a statistical technique to aggregate studies. It is part of a review framework, but is not a framework itself.

Line 64 – This is not an accurate explanation of random/mixed effects meta-analysis. This is why moderators are used. Random effects are used to extrapolate results outside the studies that are summarized. See JSS paper introducing package metafor.

Line 67 – is ‘knowledge’ what is being evaluated here? Meta-analysis doesn’t evaluate knowledge. It aggregates study results and identifies sources of variation.

Line 68: objective 1 is long, complex and likely multiple objectives

Line 72 – isn’t obj 3 listed as part of obj 1?

Line 79 - this first sentence seems like it belongs in the last paragraph of the Intro vs first paragraph of methods.. and then delete the "To achieve this goal," and start methods with "We used a response ratio approach..."

Line 83 - don't think you need the (hereafter...) clause in this sentence.. I'd assume the meta-analysis approach be referred to as the meta-analysis already... ? That said, I'm not sure saying you used a meta-analysis approach to control for variability among studies is an appropriate explanation of meta-analysis – as it doesn’t ‘control’ for variability so much as quantify variability and estimate efficacy.

Line 85 - similar to comment earlier.. I'm unclear about what is specifically meant by "tested multiple statistical models".. did you compare models using a model selection approach? Or another way of testing GOF?

Line 87 - not sure what you mean here by absolute cut-in speed and change in cut-in speed.. and were these in competing models?

Line 88 - unclear how best models were selected and the framework for model comparison ?

Line 89 - were there other covariates? I think it would be very helpful to describe the model structure and set of covariates or predictor variables and how they relate to the hypotheses being tested.

Line 108&111&117 – need to clarify/specify how the n=43 + n=22 ends up with final n=36? *see comments on Fig 1.

Line 119-120 – was a defined correlation matrix used here? See: Gleser, L. J., & Olkin, I. (2009). Stochastically dependent effect sizes. In H. Cooper, L. V. Hedges, & J. C. Valentine (Eds.), The handbook of research synthesis and meta-analysis (2nd ed., pp. 357–376). New York: Russell Sage Foundation. See also metafor documentation/examples on the package website

Line 121 – given that variability among studies is high and greatly dependent on sample size, this approach is dubious

No mention of weighting, which is highly important. Problematic to give the same weight to a study of 1 vs 100 turbines.

Table 1: which studies used global average SE? What was the global average SE and how was it calculated? Confused re the Source column and the footnotes re source? – what’s the difference between footnote #2 and listing AWWIC in the Source column (same re CanWEA and footnote 3). Add sample size of number of turbines in each study to table for comparison. What were the assumptions that were violated to warrant exclusion for Talbot Wind (footnote 1)?

Line 148-156 – Unclear. If I understand correctly, then this approach assumes fatalities are distributed evenly during the night, which is likely not the case. Were moderators used to account for study length and seasonal timing (spring vs fall) ?

Line 160/161 – effect size usually takes into account a measure of variability, how was this included?

Line161/162 – effect size does not account for changes in studies. It is simply a common metric. Moderators can be used to account for differences among studies

Line 165/168 – this seems to contradict an early statement in Methods on line 87 re evaluating models based on change in both relative and absolute cut-in speed?

Line 173/175 – fatality estimates are not normally distributed or even close to normal. They are bounded by zero and therefore using a symmetrical/normal distribution to estimate the variance around the fatality rate is flawed. A different error distribution needs to be specified and simply stating that normal approximation was the best available strategy is insufficient.

Line 177 – unclear when the delta method used or was the CI SE calc previously described was used.

Line 179/181 – description appears to be using an outdated and flawed approach and does not reflect current available advice/vignettes from the metafor package that notes that studies with shared controls and multiple treatment need a defined correlation matrix

Line 182 – mean SE likely flawed given SE is correlated with mean fatality rate and sample size

Line 186 – unclear the value of binning into 3 groups vs the linear approach? Was there a different question being asked here or were there problems fitting with using delta as continuous variable? In general, if you can avoid arbitrary binning, you should, unless there is a specific question that requires it.

Line 195-ish – were studies weighted by their sample size at all?

Line 199 – needs clearer explanation (in supplemental is fine) re what constituted high leverage and justified removal.

Line 202 – What level of ecoregions were used? The names/geographic descriptions given here do not correspond to EPA ecoregions. Eco regions are based on ecological delineations not geographic descriptors like Northeast, East, etc. Unclear how geographic regions were delineated and studies assigned to those groups.

Line 205 – “There were” [data are plural]. How did you determine a ‘lack of data for site dependencies’? This is important and should be included. Stating ‘lack of data’ without a test result is insufficient

Line 208 – Is there a list of a priori candidate models used for AIC selection?* The statement re model weights being calculated for each model type separately is confusing given that model weights are calculated based on the relative likelihood of a given model compared to the set of models. Candidate model lists, their model structure, and model selection criteria (deltaAIC, weights, etc) should be provided*. Oh you meant for categorical vs. linear types… why separately? *I found it in the Supplemental. Suggest moving into main document. Given the description of how the analysis was conducted, I’m unclear why categorical vs linear are treated as two separate candidate model sets? Couldn’t you include the categorical binning factor as a term in the model and test within the same context of the same candidate model set and determine whether that fit the data best? As presented, it leads to results section that is somewhat repetitious and unclear without much additional value in terms of understanding the core questions being asked. After examining the figures, I think the categorical approach should just be scrapped. What basis is there for binning between Category 1 and 2? Unclear what value the categorical approach provides.

Were there null models used in the model selection set? Looks like all models have Δcut-in ?

Report either AIC or AICc but not both in table.

Supplemental material mentions that the parameter estimates are provided in Table A3 but no such Table exists in the Supplemental document.

Line 212 – what is a meta-analysis scale?

Line 217 – B1 and B2 are not subsequent reductions; they are directly compared to B0 since the decrease at B2 is not B1+B2 it is simply B2

line 224 – uniform distribution is not appropriate; why not random based on actual distribution?

line 266 – a negative binomial likely a superior distribution to Poisson. Was that tested?

line 266-278 – how were the distributions selected here chosen? Does this align with the statistical guidance on fatality estimation available from the Fatality Estimator from USGS that specifies distributions for carcass persistence and searcher efficiency?

Line 280 – insufficient information of data used

Line 324 – 16 projects contradicts the earlier statement that site effects could not be calculated due to insufficient data. General random effects (how you could account for site) suggests only 4-5 levels are needed.

Line 327 – 329 – I’m still confused as to the value of comparing the binned vs continuous modeling

Line 330 – is the forest plot a figure? How would this show publication bias?

Lines 323-371 – generally the model comparison methods and results needs better description and presentation. Table A2 should be in results section. See comments earlier regarding confusion on why presenting categorical vs continuous as two separate model sets. Confusing in lines 337-338 reporting on parameter significance of rotor diameter and geographic region as these were not in either best-fit model. Table A3 giving parameter estimates mentioned in Supplemental document is missing

Line 442 – The beginning of the Discussion should emphasize the main findings of the study vs recap older findings.

Line 446 – why report the estimate from Category 1 bin in the Categorical analysis vs the results from the continuous linear analysis? The model selection approach of separating into two different analyses of categorical vs linear but then emphasizing the results of the categorical here is confusing and somewhat misleading. Why not present the linear estimate? If the continuous linear response to change in cut-in speed fits better than the categorical, even if the slope of the coefficient is only marginally significant (with low power), then that’s an indicator that there is a ‘marginally significant’ linear relationship with fatality reduction and cut-in speed? I’m not convinced the comparison of the coefficient estimates in the categorical bins approach is a robust or reasonable way to analyze/interpret.

Line 456-7 – report the result of Whitby et al vs stating it corroborates.. hard to tell what Whitby et al found or how it’s relevant here with the way it is worded. Not clear what is meant by “how volatile these results can be with sample sizes are low” – were results from Whitby et al volatile or the results here? Or they disagree? Unclear.

Line 490 – why unlikely? Doesn’t seem logical (or based on any of the data) to claim the relationship between fatality reduction and absolute cut-in speed would not be linearly related.

Line 491 – control cut-in speed appears in the top models for both model sets! Why use AIC model selection if you are going to ignore the results? Also not clear how refs 5 and 39 are appropriate here as neither of those studies address efficacy of curtailment (or spatial variation in efficacy of curtailment).

Line 492-502 – the writing and logic are muddled in this paragraph – seems in part to be justifying RR approach (not sure why that needs to be done in Discussion; and the logic is inconsistent on this point between lines 492 and 493) but then the rest of the paragraph is about variation in fatality rates and need for more research on curtailment. This paragraph needs a clearer topic and supporting points to make sense.

Line 545 – 554 – this seems irrelevant to this paper and not evaluated or discussed in enough detail or in the context of the findings to be included.

Line 558 – this is different than the result presented at the top of the Discussion section or in the Abstract.

Line 563 – not clear what “this is the case” is referring to here or what “that result” – which result from Whitby et al?

Line 566-568 – unclear what is meant here

Line 568 – the development of the efficacy of “smart” curtailment isn’t a conclusion of the work as presented.

Line 575 – what is meant by adaptive management framework in this context?

Figure 1: how does n=26 from 17 project sites turn into n=36?

Figure 2: change x-lab to Δ cut-in speed. Figure legend says relationship between bat fatality, but it’s the fatality ratio plotted on the y-axis. Explanation of how/why Talbot Wind was excluded as an ‘outlier’ needs more justification. FYI - It appears to have one of the lowest uncertainty estimates, yet is removed because of lower fatality ratio?

Figure 4: insert “speed” on x-label. Are all the grid lines necessary? Seems cluttered. See comments above re the value of doing both categorial vs linear version of analysis. Looking now at this plot, there doesn’t seem to be a logical binning between category 1 and 2. I suggest scrapping the categorical approach to the analysis. If you can justify a reason to keep it, then you should color code the points on this graph to what category they are placed in. The arbitrary cut-off at 1.4 m/s without clear bins between Category 1 and 2 will be pretty obvious when you do that, I suspect.

Figure 5: not clear what “current knowledge” means in the Scenario legend.

6. PLOS authors have the option to publish the peer review history of their article (what does this mean?). If published, this will include your full peer review and any attached files.

Reviewer #1: No

Reviewer #2: No

---

## [Author Response · Author response to Decision Letter 0]

28 Oct 2021

Reviewer 1: Line 1: sorry, I am not a native english speaker so may be my comment could appear not relevant. The word "blanket" before "curtailment strategies" (a common key word in this area) is a little vague to me, may be you could use a more explicit word for non native speakers and non specialists?

We have removed the term “blanket” throughout the manuscript, except in one location where we present it as an alternative term for curtailment or operational curtailment (as those terms are used in this paper).

Reviewer 2: Title + Line 9 - suggest removing the term 'blanket'. There's no additional meaning/value added by the word blanket in this context. I'm aware that is a jargon/term used by wind industry to try and distinguish from 'smart' curtailment but that distinction is arbitrary and I suspect designed to make it sound like curtailment strategies based on wind speeds are insufficiently sophisticated.

See response to Comment #1, above.

Reviewer 1: Line 4:The reader don't know where, to complete? 

The corresponding address has been added.

Reviewer 2: Line 9 - same comment as above – suggest removing “Blanket”.

See response to Comment #1, above.

Reviewer 2: Line 9 - 'accepted' -- really? Accepted by whom? Certainly not universally accepted by the wind industry.

The term "accepted" has been deleted from this sentence.

Reviewer 1: Line 11: I suggest "can effectively reduces" 

We have accepted this suggested edit.

Reviewer 2: Line 11 - reduces impacts? or reduces fatalities?

We have accepted this suggested edit and are being more specific with ‘fatalities’.

Reviewer 1: Line 12: may be state why there is a link between cut-in speed (based on wind speed? temperature? others?) and collision risk?

We added a brief explanation (“bat activity is higher during periods with lower wind speeds,”) to the preceding sentence. 

Reviewer 2: Line 14-15 - What is meant by 'tested multiple statistical models' ? did you use statistical models to test hypotheses? Or use multiple statistical models to explore relationships? Testing models seems odd wording and makes the intent unclear.

We changed "tested" to "used".

Reviewer 1: Line 20: I think in the abstract you could summarize what kind of scenarios were included in your study

We have added a sentence to mention this in general terms; however, we felt that defining the five specific scenarios required a more detailed explanation than was fully appropriate for the abstract.

Reviewer 2: Line 16-20 - seems a bit detailed for an abstract

We have generalized these points to cut down on the abstract complexity.

Reviewer 2: Line 22 – response ratio is the way to measure the effect size, not an ‘approach’

We have adjusted this wording.

Reviewer 2: Line 24 –the power analysis shows power is low if relationship is <50% (which would be the case with small increases in cut-in speed), so is this interpretation minimizing the existing evidence?

We have addressed this point in added text in the following sentence.

Reviewer 1: Line 25: which could explain the absence of association between increasing cut-in speed and fatalities?

Yes – we made some edits to state this more explicitly in the text.

Reviewer 2: Line 33 –explanation is not the right word here. Turbine attraction is a hypothesis that has not been proven, but is what we think may be happening (without knowing the ‘why’). The explanation for fatalities is that blades hit and kill bats.

We have changed the word “explanation” to “hypothesis”.

Reviewer 2: Line 38 –specify conservation concern is for populations due to cumulative impacts

We have rephrased this sentence slightly to clarify our point.

Reviewer 2: Line 39 - not sure what is meant by 'accepted' ; also the citation here doesn’t seem to relate to the sentence as written re operational minimization as ‘accepted’ tactic?

Reworded sentence to remove “accepted”.

Reviewer 2: Line 42 – reduce blade spinning rates below cut-in speed. The way this is written reads like blade spinning is reduced all the time vs below specified cut-in speed.

Due to other edits to this paragraph, this sentence has been deleted altogether, as it is no longer necessary.

Reviewer 2: Line 43- replace ‘still’ with ‘can’ spin.

We have accepted this suggested edit.

Reviewer 2: Line 43/44 – make feathering a new sentence and define it more clearly for clarity.

Feathering is not the main focus of the paper, and this would go into more detail than we think is appropriate. We had difficulties even getting data on which studies were feathered in this analysis, and we don’t want it to be the focus of this manuscript.

Reviewer 2: Line 48 – this sentence is misleading and incorrect as written. The AWWI summary report cited does not report fatalities associated with turbines operating under curtailment regimes (see Page 9 of AWWI summary report, criteria 2: Turbines operated at normal procedures (e.g., studies conducted while turbines were operating under a curtailment regime were not included)). The statement “a great deal of variability has been reported in the level of fatality reduction achieved by curtailment” needs clearer attribution and substantiation with data/appropriate reference.

We have restructured this sentence to clarify what information is from the cited reference and added a second citation. 

Reviewer 2: Line 49 - replace 'impacts' with 'fatalities'.

We have accepted this suggestion.

Reviewer 2: Line 49 - the wording here is verbatim as in the abstract .. what does 'exact nature of the relationship' mean? Seems like this is written to justify the work but I'm not sure the point of a meta-analysis is to get at the 'exact nature' ... isn't it more to determine general patterns or estimate average/mean effects from site-specific studies?

We have revised this sentence to clarify our point—that we want to specifically understand the role of changes in cut-in speed in determining bat fatalities.

Reviewer 2: Line 51 - the fact that you had to put 'blanket' in quotes supports my earlier comment to just delete the term. It's jargon without much useful purpose. What does the leading preposition phrase, "For this study..." mean here in this context? Doesn't curtailment have both operational and financial implications for wind facility operators outside the context of this study? The cited study does not use the word “blanket”

See our response to the first suggestion.

Reviewer 2: Line 53 - the phrase 'exact nature' stands out to me here as well.. it's very rare we can estimate or determine the exact nature of anything.. almost all relationships have some uncertainty associated with them. This sentence could be simplified by reducing to, "The trade-offs between turbine energy productions and bat fatality minimization are poorly understood."

We have accepted this suggestion.

Reviewer 2: Line 55 - the word "Still" doesn't work here. Delete. "this type of assessment" - isn't clear what type of assessment you are referring to.

We have made edits to this sentence to address the reviewer's comments and clarify our meaning. 

Reviewer 2: Line 57 - delete 'blanket' (here and elsewhere used in the manuscript)

See response to Comment #1, above.

Reviewer 2: Line 57-59 – this confuses the reasons for raising cut-in speed to 6.9m/s. The reality is that curtailment, especially high wind speed curtailment, is done as an avoidance measure to prevent fatalities of endangered species. 6.9 m/s is not often implemented as a minimization method for non-listed species.

We have deleted part of this sentence in response to the reviewer’s comment.

Reviewer 2: Line 62 – meta-analysis is a statistical technique to aggregate studies. It is part of a review framework, but is not a framework itself.

We are describing it as a statistical framework in this case and have made edits to the text to clarify our description of the technique.

Reviewer 2: Line 64 – This is not an accurate explanation of random/mixed effects meta-analysis. This is why moderators are used. Random effects are used to extrapolate results outside the studies that are summarized. See JSS paper introducing package metafor.

Thank you for pointing this out. We have modified the sentence to clarify the use of random effects meta-analysis to provide inference outside of the studies included in the analysis. Though note we have changed the specification of the random effects in the revised version of the analysis, so the interpretation is different than before.

Reviewer 2: Line 67 – is ‘knowledge’ what is being evaluated here? Meta-analysis doesn’t evaluate knowledge. It aggregates study results and identifies sources of variation.

We have deleted this phrase in response to the reviewer’s comment.

Reviewer 2: Line 68: objective 1 is long, complex and likely multiple objectives

Other characteristics such as geography and turbine dimensions are not central to our objective – rather, they are potential confounding factors in our analysis. Thus, we have deleted mention of them from our objective statement to simplify the description.

Reviewer 2: Line 72 – isn’t obj 3 listed as part of obj 1?

We have clarified the wording of objective 1 (see our response to the preceding comment).

Reviewer 1: Line 78: In the first two paragraph you could state that each procedures of this summary are developed in specific parts below, to avoid readers to anticipate specific questions about statistics

We added a sentence noting that methods to address each of the study’s three objectives are detailed below.

Reviewer 2: Line 79 - this first sentence seems like it belongs in the last paragraph of the Intro vs first paragraph of methods.. and then delete the "To achieve this goal," and start methods with "We used a response ratio approach..."

The first sentence is stated in slightly different terms in the last paragraph of the introduction already, so we have deleted it here. 

Reviewer 2: Line 83 - don't think you need the (hereafter...) clause in this sentence.. I'd assume the meta-analysis approach be referred to as the meta-analysis already... ? That said, I'm not sure saying you used a meta-analysis approach to control for variability among studies is an appropriate explanation of meta-analysis – as it doesn’t ‘control’ for variability so much as quantify variability and estimate efficacy.

We have modified the sentence. We are using the italicized meta-analysis to distinguish from the two other aspects of the study (meta-analysis power analysis, fatality estimation power analysis) as we have sections on each later in the methods. Early reviews of the manuscript found these distinctions important for clarity. We have also changed this to indicate that we are incorporating rather than controlling for variability among studies.

Reviewer 2: Line 85 - similar to comment earlier.. I'm unclear about what is specifically meant by "tested multiple statistical models".. did you compare models using a model selection approach? Or another way of testing GOF? 

We have modified this sentence to clarify that we used a model selection approach.

Reviewer 2: Line 87 - not sure what you mean here by absolute cut-in speed and change in cut-in speed.. and were these in competing models?

We have modified this sentence to clarify that we used control cut-in speed in order to allow for absolute cut-in to influence the model.

Reviewer 2: Line 88 - unclear how best models were selected and the framework for model comparison?

We have clarified above that we used a model selection framework for the analysis. We go into greater depth into the models and selection process later in the methods.

Reviewer 2: Line 89 - were there other covariates? I think it would be very helpful to describe the model structure and set of covariates or predictor variables and how they relate to the hypotheses being tested.

We go into greater detail in the section below on the model structure, predictors, etc. We have added an indication of this at the end of the paragraph, so the reader knows there is more information coming.

Reviewer 1: Line 98-101: using simulated data?

Yes, thank you, we have clarified this.

Reviewer 2: Line 108&111&117 – need to clarify/specify how the n=43 + n=22 ends up with final n=36? *see comments on Fig 1.

We have modified both in the text and Figure 1 to ensure the distinction between a project, site, and study is clear.

Reviewer 1: Line 114: unclear to me, do you mean only studies with treatment and control turbines, and thus that each turbine did not share both randomized treatment and control in time?

There is a great deal of variation in the study design of these curtailment studies, and whether controls and treatments were conducted at the same time on different turbines, and whether the turbines used for controls and treatments changed over time. Given this high level of variation, we included all of these studies. The key here, which we have clarified be deleting “of turbines,” is that the study needed to have both a treatment and a control.

Reviewer 2: Line 108&111&117 – need to clarify/specify how the n=43 + n=22 ends up with final n=36? *see comments on Fig 1.

We have modified both in the text and Figure 1 to ensure the distinction between a project, site, and study is clear. Note that we have slightly altered the figure from PRISMA format to achieve our objective with it. However, all the required information is present.

Reviewer 2: Line 119-120 – was a defined correlation matrix used here? See: Gleser, L. J., & Olkin, I. (2009). Stochastically dependent effect sizes. In H. Cooper, L. V. Hedges, & J. C. Valentine (Eds.), The handbook of research synthesis and meta-analysis (2nd ed., pp. 357–376). New York: Russell Sage Foundation. See also metafor documentation/examples on the package website

Thank you for pointing out this option in the metaphor package. We have since implemented this approach using a covariance matrix of standard errors (which is discussed below in the meta-analysis section rather than here in the Data Inclusion section).

Reviewer 2: Line 121 – given that variability among studies is high and greatly dependent on sample size, this approach is dubious. No mention of weighting, which is highly important. Problematic to give the same weight to a study of 1 vs 100 turbines.

We have deleted this here in favor of a larger discussion of methods in the Meta-analysis section below. We have since modified our handling of studies with missing standard error estimates and have also gone into greater depth into weighting. See our comment two suggestions below for more details.

Reviewer 2: Figure 1: how does n=26 from 17 project sites turn into n=36?

We have clarified both in the figure and text that multiple experimental treatments at a particular project resulted in 36 control-treatment pair studies. See our above comment changes to Fig. 1.

Reviewer 2: Table 1: which studies used global average SE? What was the global average SE and how was it calculated? Confused re the Source column and the footnotes re source? – what’s the difference between footnote #2 and listing AWWIC in the Source column (same re CanWEA and footnote 3). Add sample size of number of turbines in each study to table for comparison. What were the assumptions that were violated to warrant exclusion for Talbot Wind (footnote 1)?

We have now modified our standard error calculations for those studies lacking data (see Meta-analysis section in the Methods). We are now using multiple imputation to estimate the unknown standard errors and account for uncertainty in the true values. Those that have been estimated are indicated by asterisks. We have removed the footnotes and included relevant information in the table caption. Finally, we have included Talbot Wind in our analysis, though there is some evidence that it is influential (more information on this is included in the results/discussion).

Reviewer 1: Line 145-147: you could add this information in the table 1

Thank you for the suggestion – we have chosen to show only information in the table that was used in the analysis, and as such, we have decided not to include the estimator used in the table.

Reviewer 1: Line 149-151: it is quite difficult to see the structure of the database and at what scale bat fatalities are summarized, for each survey date inside each study? or at the study scale? if you used the scale of each survey date I do not understand why you need to apply this conversion

We have clarified in an above sentence that each study included one fatality estimate (per turbine) for the entire study for the treatment and control. We do not have finer-scale estimates of fatality (by survey), which is why we use this approach of assessing fatalities per turbine nights and hours.

Reviewer 2: Line 148-156 – Unclear. If I understand correctly, then this approach assumes fatalities are distributed evenly during the night, which is likely not the case. Were moderators used to account for study length and seasonal timing (spring vs fall) ?

We do not assume that fatalities are distributed evenly over space or time in the meta-analysis. Rather we use turbine-hours to scale the fatality rates across studies. While it is expected for fatality rates to vary over time, each treatment and control were paired to minimize those differences. By looking at the response ratio between a treatment and a control, we can reasonably assume that temporal effects are affecting each fatality assessment similarly. We added additional text to clarify this point.

Reviewer 2: Line 160/161 – effect size usually takes into account a measure of variability, how was this included?

When using a response ratio approach, an estimate of variability is not directly included in the effect size estimate. We’ve included the reference to Lajeunesse (2011) here to clarify this point. However, variability in the effect size estimate is accounted for in the weighting process described later on.

Reviewer 2: Line161/162 – effect size does not account for changes in studies. It is simply a common metric. Moderators can be used to account for differences among studies

Here, we meant that the response ratio helps control for differences in study designs that are difficult to quantify. Upon reflection, this seems evident without our additional description, so we’ve removed this sentence to avoid confusion.

Reviewer 2: Line 165/168 – this seems to contradict an early statement in Methods on line 87 re evaluating models based on change in both relative and absolute cut-in speed?

We have modified this sentence to clarify that the focus of the study is on the relative change in cut in, but we use control cut-in speed as a moderator.

Reviewer 2: Line 173/175 – fatality estimates are not normally distributed or even close to normal. They are bounded by zero and therefore using a symmetrical/normal distribution to estimate the variance around the fatality rate is flawed. A different error distribution needs to be specified and simply stating that normal approximation was the best available strategy is insufficient.

While it is true that the domain of fatality estimates is positive, we found that the confidence intervals of these estimates were symmetrical once log-transformed, which indicated that a normal approximation was appropriate in this situation. Given that the parameters of these models are estimated based on a Gaussian assumption with a log-link, we think that this approach is reasonable for estimating fatality estimate standard errors.

Reviewer 2: Line 177 – unclear when the delta method used or was the CI SE calc previously described was used.

We clarified this description. The delta method was used to create a combined standard error estimate for each treatment/control comparison. The new text clarifies that the delta method was used for uncertainty estimates in the derived fatality ratio.

Reviewer 2: Line 179/181 – description appears to be using an outdated and flawed approach and does not reflect current available advice/vignettes from the metafor package that notes that studies with shared controls and multiple treatment need a defined correlation matrix

Thank you for bringing this to our attention, this option in the metafor package was not available when we initially ran the analysis, and we have now updated our methods to include a variance-covariance matrix across studies to account for shared controls.

Reviewer 2: Line 182 – mean SE likely flawed given SE is correlated with mean fatality rate and sample size

We agree that using mean SE is a simplistic approach that doesn't account for the uncertainty in the unknown values properly. We are now using a multiple imputation approach, as suggested by Ellington et al. 2015 and Jakobsen et al. 2017 for missing information in meta-analyses. Please see the revised methods section (or the introduction to our comments) to read more about the updated approach. 

Reviewer 2: Line 186 – unclear the value of binning into 3 groups vs the linear approach? Was there a different question being asked here or were there problems fitting with using delta as continuous variable? In general, if you can avoid arbitrary binning, you should, unless there is a specific question that requires it.

We have clarified that we chose this approach to examine delta cut-in as both a continuous and categorical variable to examine the potential for both linear and non-linear relationships with bat fatality. We agree that arbitrary bins can be problematic, which is why we include information on our bin selection in the supplemental information. In sum, we used this approach because it allowed us to explore non-linear relationships between cut-in speed changes and fatality reductions without specific assumptions of the form.

Reviewer 2: Line 195-ish – were studies weighted by their sample size at all?

Yes, they were weighted by their variance decoupled SEs in the previous version of the analysis (as described below in the submitted version of the manuscript). In the revised version of the analysis, we use the fatality ratio variance-covariance matrix for weighting.

Reviewer 2: Line 199 – needs clearer explanation (in supplemental is fine) re what constituted high leverage and justified removal.

After further investigation, we have made the decision to include the Talbot Wind project in our analysis but do discuss its influence in the results and discussion, as well as provide supplemental information on how the results would change if we removed this data point.

Reviewer 2: Line 202 – What level of ecoregions were used? The names/geographic descriptions given here do not correspond to EPA ecoregions. Eco regions are based on ecological delineations not geographic descriptors like Northeast, East, etc. Unclear how geographic regions were delineated and studies assigned to those groups.

We have added additional information on the ecoregion level and groupings used.

Reviewer 2: Line 205 – “There were” [data are plural]. How did you determine a ‘lack of data for site dependencies’? This is important and should be included. Stating ‘lack of data’ without a test result is insufficient

We have significantly reworked the process by which we handled site dependencies in the data. After revising the approach to handling shared controls among studies, we were able to handle the site dependencies through a random effect. We were able to solve issues with the random effect estimation process that we had previously.

Reviewer 2: Line 208 – Is there a list of a priori candidate models used for AIC selection?* The statement re model weights being calculated for each model type separately is confusing given that model weights are calculated based on the relative likelihood of a given model compared to the set of models. Candidate model lists, their model structure, and model selection criteria (deltaAIC, weights, etc) should be provided*. Oh you meant for categorical vs. linear types… why separately? *I found it in the Supplemental. Suggest moving into main document. Given the description of how the analysis was conducted, I’m unclear why categorical vs linear are treated as two separate candidate model sets? Couldn’t you include the categorical binning factor as a term in the model and test within the same context of the same candidate model set and determine whether that fit the data best? As presented, it leads to results section that is somewhat repetitious and unclear without much additional value in terms of understanding the core questions being asked. After examining the figures, I think the categorical approach should just be scrapped. What basis is there for binning between Category 1 and 2? Unclear what value the categorical approach provides.

We have included both linear and categorical models in a single model selection framework as suggested and have moved the AICc table into the main body of the manuscript for clarity. All the models, including base models that only include site and control cut-in speed, were described for each selection. We continue to see value in the categorical approach as it allows us to contrast linear and non-linear relationships between changes in cut-in speed and fatality reductions. Now that we showing the contrast in AIC, it allows us to compare them directly to help determine their relative value.

Reviewer 2: Line 208 -Were there null models used in the model selection set? Looks like all models have Δcut-in ?

This was clarified in our response to the previous comment.

Reviewer 2: Line 208 - Report either AIC or AICc but not both in table

AICc is listed in the revised table as noted in our response to the comment above.

Reviewer 2: Line 208Supplemental material mentions that the parameter estimates are provided in Table A3 but no such Table exists in the Supplemental document.

This was a mistake that we corrected in the recent version. We moved all relevant information on parameter estimates from the supplemental information as well as any information on model selection (outside of determining the appropriate categories for the categorical model).

Reviewer 2: Line 212 – what is a meta-analysis scale?

This line was clarified to mean a meta-analysis power analysis.

Reviewer 2: Line 217 – B1 and B2 are not subsequent reductions; they are directly compared to B0 since the decrease at B2 is not B1+B2 it is simply B2

Correct, we have changed the language here to make this clearer.

Reviewer 1: Lines 216-218: it is quite difficult for me to understand how did you simulated fatility reduction: from models tested above? but in case of non significant difference how did you assess the power generally expressed as the % of significant p-values?

The fatality reductions were simulated in our estimates of the response ratio and how it changed with cut-in speed. Essentially, we set the response ratio a priori, ran the analysis many times, the aggregated the results to estimate power. We clarified the description of this method in the text.

Reviewer 2: line 224 – uniform distribution is not appropriate; why not random based on actual distribution?

We agree that the uniform distribution was not ideal for this approach. In the revised analysis, standard errors were simulated using a gamma distribution based on the mean and standard error of the known standard errors.

Reviewer 1: Lines 244-251: ok nice this part is very clear 

We made a few small changes here to update this section for the methods, but thank you for the comment.

Reviewer 1: Table 2: B0, B1 and B2- may be recall here what does it means

Thank you for the suggestion. We have added a reminder in the table legend.

Reviewer 1: Line 261: I am not sure to well understand, your primary fatalities data used for simulations are already corrected data according to survival rate? Is it quite circular to then test the relationship between fatalities and persistence?

In the data simulation approach, we set the fatality rates of the two treatment groups, the detection probability, and the carcasses persistent rates. Then we use each of those values to estimate the number of detections we would see for a given simulation. It's not circular logic, but rather a consistency in the values that we used to test this experiment. Given that we know the true parameters, we create data using the same tools that GenEst would analyze it. Thus, we can assess the efficacy of GenEst in differentiating treatment groups, assuming fatality data are generated the way it assumes. 

Reviewer 2: line 266 – a negative binomial likely a superior distribution to Poisson. Was that tested?

A negative binomial model was not tested in this case. A negative binomial distribution can often describe real-world ecological data better, particularly those that are overdispersed relative to a Poisson distribution. However, the GenEst models are mixture models that have other methods for incorporating overdispersion into the estimates via observation processes like detection probability or carcasses persistence. Given it is unclear which type of distributional choice is most in line with fatality estimation studies, we choose the simpler model to test. This choice would be an interesting question to look at in future studies. We made some additions to this section to clarify our process.

Reviewer 1: Lines 269-270: may be refer to literature to support this choice

An exponential model is typically used to estimate survival rates (in this case, carcass persistence). The original citing of the GenEst package provides the needed reference here.

Reviewer 2: line 266-278 – how were the distributions selected here chosen? Does this align with the statistical guidance on fatality estimation available from the Fatality Estimator from USGS that specifies distributions for carcass persistence and searcher efficiency?

Yes, we are using the fatality estimator developed by USGS, and our simulation approach follows the guidelines they provide.

Reviewer 2: Line 280 – insufficient information of data used

The studies we used to parameterize these models are the AWWIC studies from Table 1. We added text to clarify this point and what information we used to build these simulation scenarios.

Reviewer 1: Lines 314-315: may be replace categories with true curtailment thresholds? and explicit underlying % decreases

We have added the equivalent percent decrease here in the text as well as the thresholds used to define the three categories.

Reviewer 2: Figure 2: change x-lab to Δ cut-in speed. Figure legend says relationship between bat fatality, but it’s the fatality ratio plotted on the y-axis. Explanation of how/why Talbot Wind was excluded as an ‘outlier’ needs more justification. FYI - It appears to have one of the lowest uncertainty estimates, yet is removed because of lower fatality ratio? 

We have changed the x-label as requested and clarified in the figure caption that we are showing the relationship with bat fatality ratio. After further exploration, we have also made the decision to keep the Talbot Wind project in the analysis (see Meta-analysis section below for further information on the potential influence of this point), but also describe its effect on the results of the study. This change added complexity to the discussion but was a more even-handed approach that excluded a marginally high leverage point.

Reviewer 1: Fig 2 Lines 318-319: did you accounted for pseudoreplication? sorry if you already provide the information above, the method is quite long to read and understand 

We have now accounted for this by using a variance-covariance matrix within the meta-analysis models, which accounts for the covariance among multiple treatments that use the same control (see description in the methods section).

Reviewer 2: Lines 323-371 – generally the model comparison methods and results needs better description and presentation. Table A2 should be in results section. See comments earlier regarding confusion on why presenting categorical vs continuous as two separate model sets. Confusing in lines 337-338 reporting on parameter significance of rotor diameter and geographic region as these were not in either best-fit model. Table A3 giving parameter estimates mentioned in Supplemental document is missing 

Thanks for this suggestion - we have moved the model selection table to the main manuscript and throughout the methods and results worked to clarify the model selection process. As suggested, we have also combined the categorical and continuous models into the same model selection framework. Given that the models with continuous and categorical effects of changing cut-in speeds are the top two models with <1 delta AIC, we have presented the results of both. We have also added language to clarify that rotor diameter and geographic region were not in the top models, however, we felt that it was important information to provide that even in the global models, these variables were not significant.

Reviewer 2: Line 324 – 16 projects contradicts the earlier statement that site effects could not be calculated due to insufficient data. General random effects (how you could account for site) suggests only 4-5 levels are needed.

We have removed the sentence above as we have included site as a random effect in the revised analysis.

Reviewer 2: Line 327 – 329 – I’m still confused as to the value of comparing the binned vs continuous modeling

We find value in comparing the approaches given that we cannot assume that the relationship between delta cut-in and fatality reduction is linear, and the categorical models provide a means to examine non-linearity. Given that the model selection shows the continuous and categorical delta cut-in models are included in our models, within <1 delta AIC, we have chosen to keep the comparison of the two models in the results, as we feel this adds value to the discussion.

Reviewer 2: Line 330 – is the forest plot a figure? How would this show publication bias?

Apologies, this was a mistake. We used a funnel plot to examine the potential for publication bias (not shown). Please note that we have added additional information to this section in relation to the Talbot Wind study as a potential outlier.

Reviewer 2: Figure 4: insert “speed” on x-label. Are all the grid lines necessary? Seems cluttered. See comments above re the value of doing both categorial vs linear version of analysis. Looking now at this plot, there doesn’t seem to be a logical binning between category 1 and 2. I suggest scrapping the categorical approach to the analysis. If you can justify a reason to keep it, then you should color code the points on this graph to what category they are placed in. The arbitrary cut-off at 1.4 m/s without clear bins between Category 1 and 2 will be pretty obvious when you do that, I suspect.

We have added "speed" on the x-axis label and removed grey grid lines to reduce clutter as suggested. Rather than coloring points based on category, we have added vertical lines on the graph to indicate the categories. We recognize that binning is often imperfect, however, you will find in the supplemental information (Appendix A) that we examined the sensitivity of the results to the choice of bins and found very similar results. As mentioned earlier, given our uncertainty in the nature of the relationship between bat fatalities and delta cut-in speed, as well as our model selection results, we have chosen to include both the continuous and categorical models in our results.

 Reviewer 2: Figure 5: not clear what “current knowledge” means in the Scenario legend. 

We removed the ‘current knowledge’ scenario from the analysis after our revision to the meta-analysis simulation. This made the meta-analysis power analysis simpler to explain and interpret.

Reviewer 2: Line 442 – The beginning of the Discussion should emphasize the main findings of the study vs recap older findings. We’ve revised this paragraph to incorporate the updated results.

We emphasized the findings of our study, but we do continue to provide context as to how our results compare at a high level.

Reviewer 2: Line 446 – why report the estimate from Category 1 bin in the Categorical analysis vs the results from the continuous linear analysis? The model selection approach of separating into two different analyses of categorical vs linear but then emphasizing the results of the categorical here is confusing and somewhat misleading. Why not present the linear estimate? If the continuous linear response to change in cut-in speed fits better than the categorical, even if the slope of the coefficient is only marginally significant (with low power), then that’s an indicator that there is a ‘marginally significant’ linear relationship with fatality reduction and cut-in speed? I’m not convinced the comparison of the coefficient estimates in the categorical bins approach is a robust or reasonable way to analyze/interpret.

The linear and categorical models are less than 1 AICc point apart, so we present the two best models in this situation. We have adjusted the text to discuss both models together so that it is clearer why we are discussing them both. We also discuss this further in the revised discussion section. 

Reviewer 2: Line 456-7 – report the result of Whitby et al vs stating it corroborates.. hard to tell what Whitby et al found or how it’s relevant here with the way it is worded. Not clear what is meant by “how volatile these results can be with sample sizes are low” – were results from Whitby et al volatile or the results here? Or they disagree? Unclear.

We have reported the Whitby et al. (2021) results in further detail and more clearly compared them to our own findings.

Reviewer 2: Line 490 – why unlikely? Doesn’t seem logical (or based on any of the data) to claim the relationship between fatality reduction and absolute cut-in speed would not be linearly related.

We made significant changes to this paragraph due to the changes in the results and as a result of this suggestion. We are now focusing on our ability to estimate linear vs. non-linear effects using the meta-analysis power analysis.

Reviewer 2: Line 491 – control cut-in speed appears in the top models for both model sets! Why use AIC model selection if you are going to ignore the results? Also not clear how refs 5 and 39 are appropriate here as neither of those studies address efficacy of curtailment (or spatial variation in efficacy of curtailment).

Control cut-in speed was included in all tested models to control for the potential effect. As such, the model selection results are not a particularly useful way to assess the importance of the covariate. In this case, the effect size of the parameter was small and not as important as other factors. 

Reviewer 2: Line 492-502 – the writing and logic are muddled in this paragraph – seems in part to be justifying RR approach (not sure why that needs to be done in Discussion; and the logic is inconsistent on this point between lines 492 and 493) but then the rest of the paragraph is about variation in fatality rates and need for more research on curtailment. This paragraph needs a clearer topic and supporting points to make sense.

This paragraph has been significantly reworked to respond to your comment and due to the changes in the analysis. We have taken your suggestion to have a stronger goal of the paragraph, focusing in on the lack of evidence for other factors that could affect fatality reductions.

Reviewer 1: Lines 549-552: but expected to be mainly efficient on low frequency species?

In response to the previous comment from Reviewer 2, below, we have trimmed back the amount of detail in this paragraph on alternative mitigation approaches. Thus, going into additional detail regarding efficacy for various species groups does not seem warranted here. We do say, however, that these approaches are still being evaluated.

Reviewer 2: Line 545 – 554 – this seems irrelevant to this paper and not evaluated or discussed in enough detail or in the context of the findings to be included.

This manuscript went through several rounds of peer review before it was submitted for publication, and we received several reviewer comments suggesting the importance of recognizing newer mitigation approaches that are in development. Thus, we feel that cutting this paragraph altogether would be counterproductive. However, we have trimmed the length of this section in order to avoid distracting readers from the main focus of our findings.

Reviewer 2: Line 558 – this is different than the result presented at the top of the Discussion section or in the Abstract.

We apologize for the mistake. This has been updated in the revised paper.

Reviewer 2: Line 563 – not clear what “this is the case” is referring to here or what “that result” – which result from Whitby et al?

Whitby et al. (2021) also found that changes in cut-in speed reduce bat fatalities in another meta-analysis of a partially overlapping data set. We clarified the point here and now suggest that the two studies corroborate each other.

Reviewer 2: Line 568 – the development of the efficacy of “smart” curtailment isn’t a conclusion of the work as presented.

While we now emphasize that we did not study smart curtailment directly, but we have decided to include this statement in the conclusions as a clear direction that research in this area can proceed.

Reviewer 2: Line 566-568 – unclear what is meant here

We altered the conclusions after revising the analysis and have clarified this point.

---

## [Editor Report · Decision Letter 1]

2 Nov 2021

A review of the effectiveness of operational curtailment for reducing bat fatalities at terrestrial wind farms in North America

PONE-D-21-22448R1

Dear Dr. Adams,

We’re pleased to inform you that your manuscript has been judged scientifically suitable for publication and will be formally accepted for publication once it meets all outstanding technical requirements.

Kind regards,

Ignasi Torre

Academic Editor

PLOS ONE
---

## [Editor Report · Acceptance letter]

8 Nov 2021

PONE-D-21-22448R1 

A review of the effectiveness of operational curtailment for reducing bat fatalities at terrestrial wind farms in North America 

Dear Dr. Adams:

I'm pleased to inform you that your manuscript has been deemed suitable for publication in PLOS ONE. Congratulations! Your manuscript is now with our production department. 

Kind regards, 

on behalf of

Dr. Ignasi Torre 

Academic Editor

PLOS ONE